# Ocean Color Remote Sensing of Suspended Sediments along a Continuum from Rivers to River Plumes: Concentration, Transport, Fluxes and Dynamics

**Anouck Ody** [1]**, David Doxaran** [1,*]**, Romaric Verney** [2]**, François Bourrin** [3]**, Guillaume P. Morin** [4]**, Ivane Pairaud** [5] **and Aurélien Gangloff** [6]

1   UMR 7093 CNRS/SU, Laboratoire d'Océanographie de Villefranche, 06230 Villefranche-sur-Mer, France; ody.anouck@gmail.com
2   Laboratoire de Dynamique Hydro-Sédimentaire (PDG-ODE-DYNECO-DHYSED), IFREMER, 29280 Plouzané, France; romaric.verney@ifremer.fr
3   UMR 5110 CNRS-UPVD, CEFREM, Université de Perpignan, 66860 Perpignan, France; fbourrin@univ-perp.fr
4   INRAE, Aix Marseille University, UMR RECOVER, F-13182/Pôle ECLA, 13100 Aix-en-Provence, France; guillaume.p.morin@inrae.fr
5   UMR 6523 CNRS, IFREMER, IRD, UBO, Laboratoire d'Océanographie Physique et Spatiale, 29280 Plouzané, France; ivane.pairaud@ifremer.fr
6   Service Hydrographique et Océanographique de la Marine (SHOM), 29200 Brest, France; aurelien.gangloff@shom.fr
*   Correspondence: david.doxaran@imev-mer.fr

**Abstract:** This study investigates the capability of high and medium spatial resolution ocean color satellite data to monitor the transport of suspended particulate matter (SPM) along a continuum from river to river mouth to river plume. An existing switching algorithm combining the use of green, red and near-infrared satellite wavebands was improved to retrieve SPM concentrations over the very wide range (from 1 to more than 1000 g.m$^{-3}$) encountered over such a continuum. The method was applied to time series of OLI, MSI, and MODIS satellite data. Satisfactory validation results were obtained even at the river gauging station. The river liquid discharge is not only related to the SPM concentration at the gauging station and at the river mouth, but also to the turbid plume area and SPM mass estimated within the surface of the plume. The overall results highlight the potential of combined field and ocean color satellite observations to monitor the transport and fluxes of SPM discharged by rivers into the coastal ocean.

**Keywords:** ocean color; suspended particulate matter; river mouth continuum; liquid and solid river discharges into the coastal ocean

## 1. Introduction

Climate change at regional scales impacts fluxes of terrestrial substances (e.g., suspended particulate matter (SPM) and associated particulate organic carbon (POC), nutrients, and pollutants discharged by rivers into the coastal ocean. There is growing evidence of a link between climate change and the increase (in occurrence and intensity) in extreme meteorological events such as floods and droughts (IPCC 2012), notably in the northwestern Mediterranean Sea [1,2], recently affected by extreme precipitation events and subsequent floods [3].

Many world rivers are nowadays equipped with gauging stations for the continuous monitoring of the water level and concentrations of terrestrial substances transported in suspension and solution. These measurements help estimate the freshwater and solid discharges at the land–ocean interfaces that represent river mouths (e.g., [4]). However, these gauging stations are often located up to 100 km upstream from the actual river mouths, so that a significant uncertainty remains concerning the amounts of suspended solids

actually trapped in the downstream part of rivers and the area reaching the coastal ocean. These fluxes are driven by environmental conditions (river discharge, morphology, and tidal effects) and intensive and complex flocculation processes occurring in the transition zones between fresh and oceanic waters [5,6]. Flocculation processes, which occur as soon as saline waters are reached, enhance the aggregation of cohesive particles, resulting in the formation of flocs, which rapidly sink and settle. Then, flocs can grow or break up depending on their mineralogy, organic content, concentration, and turbulence (see [7]).

Over the recent decades, scientists and environmental management agencies have dedicated a significant effort to deploy continuous high-frequency monitoring networks, providing crucial knowledge about physical and biogeochemical processes in the land–sea continuum [4,8–10]. However, in situ observations are by essence designed from a "limited number" of local measurement stations and do not provide information about the spatial continuity and variability of suspended particulate fluxes within the continuum.

Satellite data now available at high (~20 m) and medium (~300 m) spatial resolutions may be used to complementarily document the transport of SPM along these dynamic zones and contribute to numerical model validation. The processing of ocean color satellite data is now operational to monitor the transport of SPM in river plumes [11,12] and estuaries [13,14]. The general objective of the present study was to demonstrate the capability of satellite observations to study the SPM transfer and dynamics along a river to river mouth to river plume continuum, i.e., from the land to the adjacent coastal sea. Specific objectives include: to highlight the ability of high spatial and temporal resolution satellite data to monitor and better understand (i) the transport of suspended particles into the downstream part of the river, (ii) the transfer of suspended particles from the river to the river plume and (iii) the relationships between river discharge and river plume extent and SPM mass.

The method used first combines high spatial resolution satellite data (from the Landsat8-OLI (Operational Land Imager) and Sentinel2-MSI (MultiSpectral Instrument) sensors that assess the SPM concentration in the river, and high temporal resolution satellite data (from the AQUA and TERRA-MODIS (Moderate-Resolution Imaging Spectroradiometer) sensors), which cover the river mouth and plume over a large period (2013–2020). This original combined use of high and medium spatial resolution satellite observations aids the study of the river to river mouth to river plume continuum. In order to demonstrate the applicability of this global observation method to the land to sea continuum, we selected the Rhône River and Gulf of Lion system as a pilot site. We relied on in situ data from a river gauging station and another fixed station located at the river mouth (MesuRho), as well as data acquired during field campaigns within the plume area, mainly to calibrate and validate ocean color algorithms over the continuum. Finally, this method extracted relevant information on the river plume surface and SPM mass from satellite observations, which are related to the freshwater and solid river discharges documented by field data.

The Rhône River has already been used as a reference test site for the calibration of ocean color satellite algorithms [11,12]. The present study extends the work undertaken by [15], where ocean color satellite data were used to establish a robust relationship between the Rhône River discharge and metrics of the river plume in the adjacent coastal sea.

## 2. Materials and Methods

### 2.1. Study Area and Context

The Rhône River is 812 km long, originating in Switzerland and splitting into two branches, the Grand Rhône and the Petit Rhône, 3.5 km upstream from Arles (southeastern France) and about 50 km upstream from the river mouth. The Rhône River flows into the Gulf of Lion in the northwestern part of the Mediterranean Sea (Figure 1) and contributes 80% of the terrigenous material input and 95% of the freshwater discharge in this region [16–18], with the Grand Rhône River contributing to about 90% of transfer to the coastal ocean. The Rhône River freshwater discharge (Q) is characterized by an annual mean value of 1700 $m^3.s^{-1}$. Its large drainage basin, covering 97,800 $km^2$ with different climatic zones (alpine, oceanic, and Mediterranean), induces strong seasonal

and interannual variations with freshwater discharge values typically varying from less than 700 $m^3.s^{-1}$ in summer to more than 4000 $m^3.s^{-1}$ in spring and autumn, and that can exceed 10,000 $m^3.s^{-1}$ during exceptional peak flood events [18]. This generates, in adjacent coastal waters, a buoyant surface plume characterized by high loads of fine sediment particles slowly sinking in a microtidal regime area. The plume can thus extend over several tens of kilometers offshore making it easily detectable on ocean color satellite data (Figure 1). This plume shows morphological fluctuations, mainly in terms of orientation and offshore extents that both depend on wind, freshwater discharge, and coastal current conditions [19–21]. The thin surface plume (1–5 m thick) is commonly associated with a bottom nepheloid layer (BNL) formed by the sinking of particles from the surface layer and resuspension of sediments (e.g., [19,22,23]). Both usually merge near the mouth at water depths of about 20 m [23].

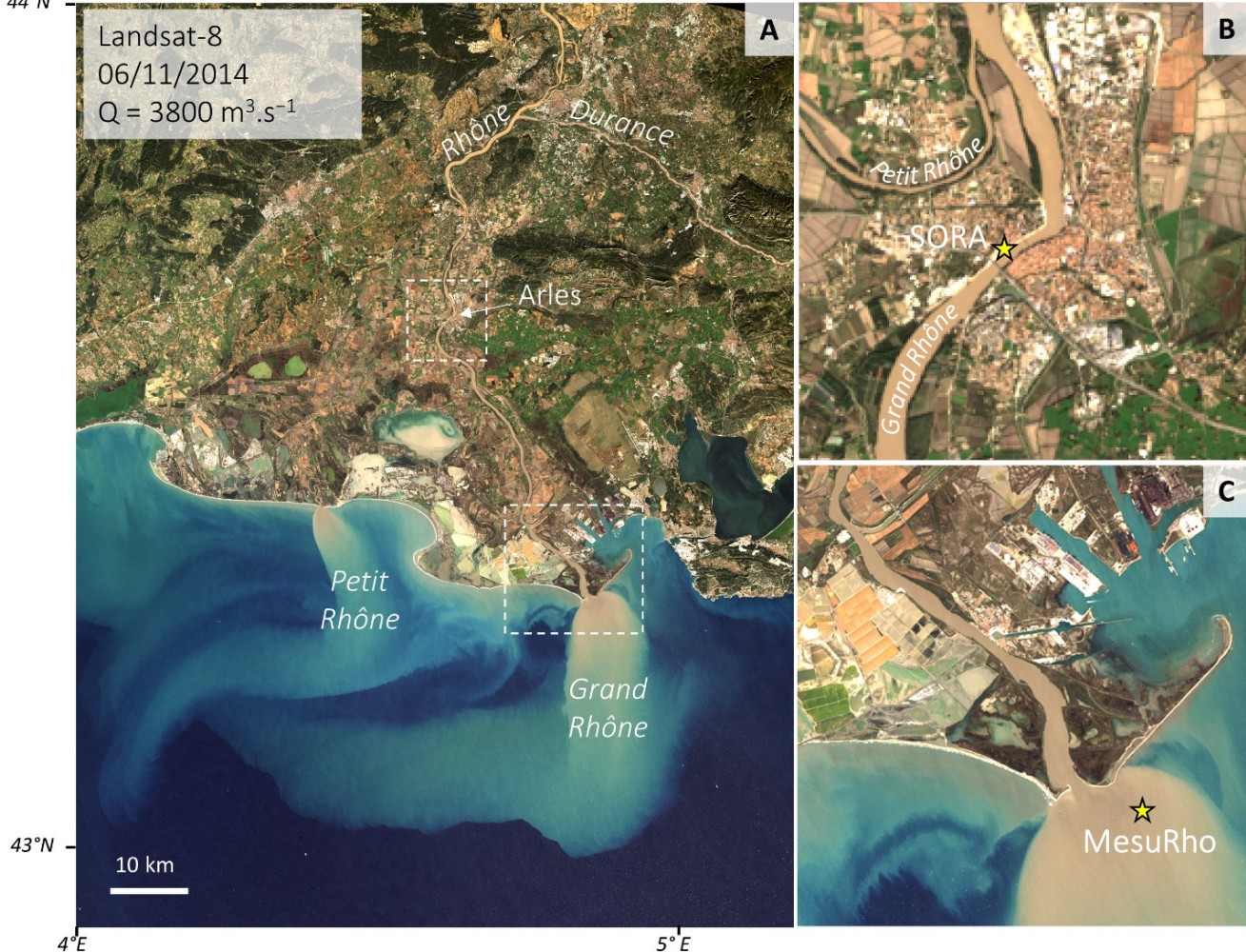

**Figure 1.** (**A**) Study area (Landsat8-OLI image, 6 November 2014). The Rhône River crosses the city of Arles and reaches the sea at about 45 km kilometers southward, forming a well-defined river plume. (**B**) The SORA station (yellow star) where the SPM concentration and associated river discharge are measured is located in the city of Arles. (**C**) The MesuRho platform is located 2 km from the river mouth, which is inside the river plume except under very strong southeastern wind conditions or very low SPM concentration periods.

The SPM concentration in the Grand Rhône River has been systematically measured in Arles since 2005 through an automatic gauging station called Station Observatoire du Rhône en Arles (SORA) ([24], Figure 1B). The measured SPM concentrations range from 1 to about 6300 g.m$^{-3}$ with a mean value of about 66 g.m$^{-3}$ (values extracted from SORA data base (2005–2020), see Section 2.2.1), leading to annual suspended solid fluxes ranging from 1 to 9 Mt [18,24]. SPM concentrations within the surface plume and at the river mouth have been measured through several field campaigns as well as through automatic measurements of SPM concentration proxies (e.g., [12]) at the fixed MesuRho monitoring platform, which is a station of the COAST-HF (Coastal OceAn observing SysTem–High Frequency) network of RI ILICO [25] and of the Rhône Sediment Observatory (OSR) program [26] located 2 km southeast of the Rhône River mouth (Figure 1C, bottom depth of 17–20 m). Measurements from subsurface water samples show SPM concentrations between ~3 g.m$^{-3}$ and more than 60 g.m$^{-3}$ [11,12]. This large SPM concentration range is associated with an heterogeneity in the suspended matter mineralogical composition due to the diversity of relative water contributions of upstream tributaries and types of flood events occurring in the large Rhône River catchment area [18].

The present study is mainly focused on the area covering the Grand Rhône River, from Arles to the river mouth (marked by the MesuRho platform) and the Grand Rhône River plume (Figure 1A). The study period (2013–2020) shows a mean SPM concentration of 78 g.m$^{-3}$ at the SORA station for a corresponding mean freshwater discharge of 1600 m$^3$.s$^{-1}$ and covers the two highest peaks of SPM concentration and freshwater discharge recorded since 2005: 6300 g.m$^{-3}$ and 6400 m$^3$.s$^{-1}$ in November 2016 (Figure 2) and 1389 g.m$^{-3}$ and 6490 m$^3$.s$^{-1}$ in December 2019. The reported particulate organic carbon (POC) content for this period (SORA station, Table 1) shows significant variations (POC-SPM ratio values from 0 to 10%) apparently inversely correlated with the freshwater discharge (Figure 2).

*2.2. Dataset*

2.2.1. In Situ Measurements from the SORA Station (Arles)

The SORA station is an automatic sampling station located at Arles on the right bank of the Grand Rhône River, 3.5 km downstream from the diffluence between the Grand Rhône and the Petit Rhône and 45 km upstream from the river mouth. Since 2005, it provides a daily sampling of water quality parameters, including SPM and POC concentrations during low and normal water stages (Q < 3000 m$^3$.s$^{-1}$), and a 4 h high frequency sampling during floods (Q > 3000 m$^3$.s$^{-1}$) [24]. Daily and hourly freshwater discharges in Arles were also made available by CNR (Compagnie Nationale du Rhône). Data were obtained from the MOOSE program (Mediterranean Oceanic Observing System for the Environment).

Daily averaged data of Grand Rhône River freshwater discharge, SPM, and POC concentrations for the 2013–2020 period were used in this study (Figure 2, Table 1). Uncertainties on SPM concentrations were estimated to $5 \times 10^{-1}$ g.m$^{-3}$ [24]. However, these uncertainties did not account for the daily (or hourly) variability of the SPM concentration. For a better comparison with instantaneous satellite-derived SPM concentrations, the uncertainty of the averaged SPM concentration measured at SORA was assumed to be the variation in SPM concentration observed within ±1 day (i.e., $max(abs(SPM_i\text{-}SPM_{i-1}; SPM_i\text{-}SPM_{i+1}; SPM_{i+1}\text{-}SPM_{i-1})))$. During high flood events, we used the SPM concentration measurement that was temporally the closest to the satellite acquisition (Table 1), and as measurement error, the maximal SPM concentration interval observed within ± 4 h.

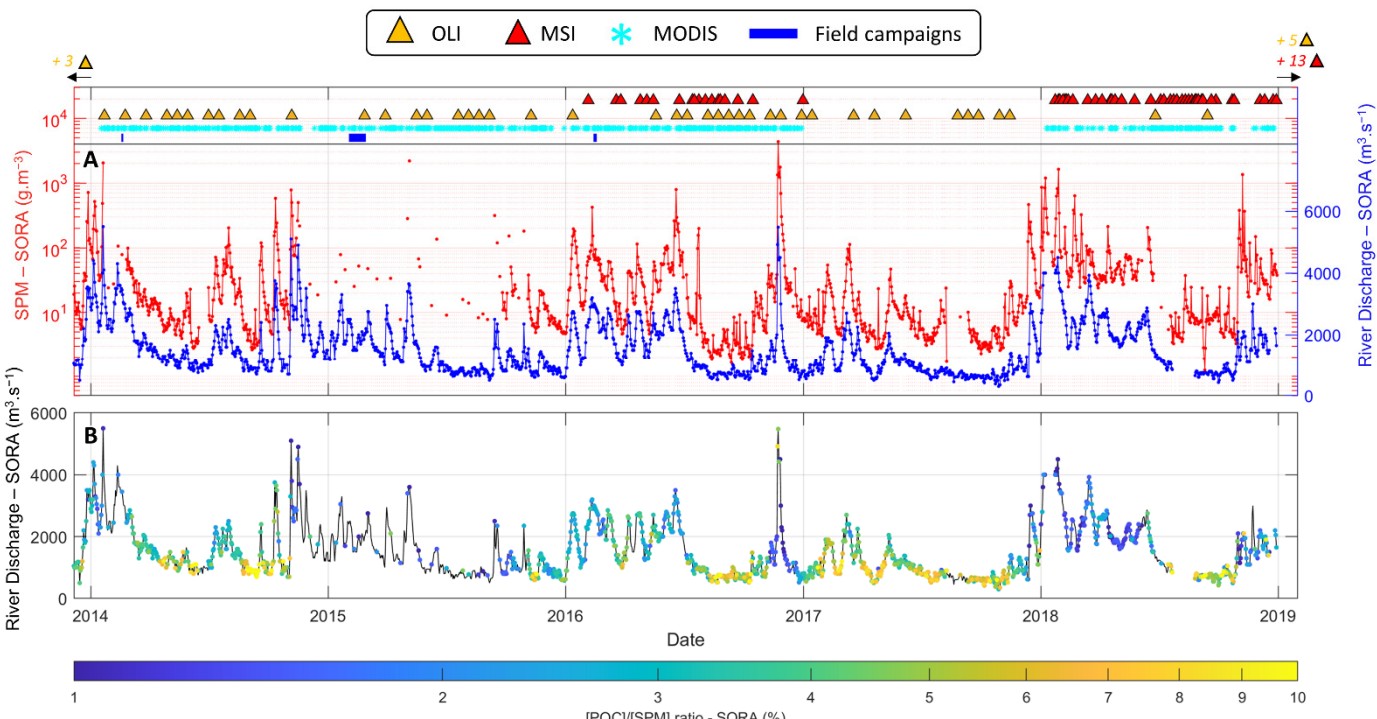

**Figure 2.** (**A**) Daily averaged in situ SPM concentration and river discharge measured at the SORA station (Arles) from 2014 to 2018. Days with cloud-free OLI (yellow triangles), MSI (red triangles), and MODIS (cyan asterisks) satellite data are indicated in the top part of the panel. The number of additional OLI and MSI images in 2013 and 2019–2020 are indicated in the top left and right corners, respectively. The field campaign periods with SPM and radiometric measurements are also indicated (blue line). (**B**) POC-SPM ratio (colored points) measured at the SORA station overlapping the Rhône River discharge (black line).

### 2.2.2. In Situ Measurements from Field Campaigns

Simultaneous radiometric and SPM concentration measurements were carried out during three campaigns in the Rhône River plume (along shore and cross shore transects) in February of 2014 (TUCPA), 2015 (PLUMRHO), and 2016 (MATUGLI).

Above-water radiometric measurements were performed with two radiance and one irradiance TriOS-RAMSES hyperspectral sensors. The two radiance sensors were used to measure the total upwelling and sky radiances, $L_t$ and $L_s$ (W.m$^{-2}$.sr$^{-1}$), respectively. They were mounted so their nadir and zenith angles were 40° (as recommended by [27,28]), and their solar relative azimuth angle was around 135° (angle actually varying from 90° to 160° depending on hour and sea-state conditions). The irradiance sensor was pointed to the zenith to measure the downwelling irradiance signal ($E_d$, W.m$^{-2}$) and was mounted on a vertical mast away from shadow effects. The sensors spanned the 350–950 nm spectral range with a spectral resolution of 3.3 nm. Data were recorded using TriOS GmbH MSDA software using the nominal calibration coefficients set in October 2014. Calibrated data for $E_d$, $L_t$, and $L_s$ were then interpolated to 1 nm intervals before processing.

Measurements last 2 to 12 min for each station. Simultaneous spectra from the three sensors were acquired every 10 s and the remote-sensing reflectance signal (Rrs, sr$^{-1}$) was computed as:

$$Rrs = \frac{L_t - \rho L_s}{E_d} \tag{1}$$

where $\rho$ is the air–water interface reflection coefficient [27]. This coefficient may vary strongly with wind speed for clear sky conditions because of reflection of brighter parts of the sky in presence of waves [27] but is approximately independent of wind speed

under overcast skies. When wind data was available, it was accounted for by switching between clear sky and cloudy sky models for $\rho$, according to the ratio $L_s$-$E_d$ at 750 nm (see Equations (23) and (24) in [28]). Otherwise, $\rho$ was taken as a constant value of 0.0256 [28]. For each set of measurements, the measured Rrs spectra were filtered and averaged. Spectra were selected during a period with stable illumination conditions and a low inclination of the Ed sensor ($<10°$) and outliers were removed before averaging [12]. The water-leaving reflectance signal (mean value and standard deviation, dimensionless) was finally computed as: $\rho_w = \pi \times$ Rrs.

During field campaigns, water samples were collected at each station, simultaneously with radiometric measurements. They were collected within the surface layer (0 to 1 m depth) using one horizontal Niskin bottle. SPM concentration was determined by filtering at low-vacuum known volumes (V in $m^3$) of seawater through pre-combusted (450 $°C$) and pre-weighed ($M_0$, in g) glass-fiber filters (Whatman GF/F, 0.7 nominal pore size), as described in [29]. Each filter was then rinsed with Milli-Q water and stored in the cold on the ship and subsequently at $-80$ $°C$ once back at the laboratory. At final processing, filters were dried for 24 h at 65 $°C$ then weighed (M, in g) under a dry atmosphere in order to obtain the SPM concentration:

$$\text{SPM} = \frac{(M - M_0)}{V} \left( \text{g.m}^{-3} \right) \tag{2}$$

### 2.2.3. Satellite Dataset

To follow the sediment transport from the Rhône River to the Rhône River plume, we combined high spatial resolution and high temporal resolution satellite data. We considered three satellite-borne sensors:

- The OLI sensor on the Landsat-8 (L8) polar-orbiting satellite platform launched in 2013. This sensor provides multispectral data with a high spatial resolution of 30 m and a temporal resolution of 16 days;
- The MSI sensors on the Sentinel-2 A (S2-A) and B (S2-B) European polar-orbiting satellite platforms launched in 2015 and 2017, respectively, which provide high spatial resolution data (10 to 60 m) and a temporal resolution at the study area latitude of 2–3 days using both satellite platforms;
- The MODIS sensors aboard the polar-orbiting Terra (MODIS-T) and Aqua (MODIS-A) satellite platforms launched in 1999 and 2002, respectively. They provide multispectral data with a revisiting time of one day at the latitude of the study area (thus 2 MODIS images per day with a ~2 h gap between them), with three spatial resolutions of 250 m, 500 m, and 1 km, depending on the spectral band.

To provide representative and statistically robust results, this study is based on large MODIS, OLI, and MSI datasets covering several years between 2013 and 2020 with a wide range of Rhône River discharge conditions and SPM concentrations measured in Arles (Figure 2).

For OLI and MSI, we downloaded cloud-free (<10%) images acquired over the study area between 2013 and 2018. Given the low temporal resolution of OLI, all cloud-free images available over this period were downloaded and processed. For MSI, only images acquired during the 2016 and 2018 years corresponding to simultaneous in situ SPM concentration measurements at the SORA station were selected. To increase the amount of data with high SPM concentration, OLI and MSI images acquired during Rhône River high turbidity periods (SPM concentration measured at SORA station > 70 g.m$^{-3}$) in 2019 and 2020 were added to the dataset. Note that the study area was covered by two successive MSI tiles switching at ~43.25°N. In order to limit the data processing time, we only kept the tile covering the Rhône River from Arles to the northern part of the river plume (latitude > 43.25°N). Orthorectified and terrain-corrected Level-1T OLI and Level-1C MSI products were obtained from the Landsat-8 portal on the ESA website (https://landsat8portal.eo.esa.int/portal/ (accessed on 22 February 2022)) and from the

Copernicus open access hub (https://scihub.copernicus.eu/ (accessed on 22 February 2022)), respectively. Both were processed using the ACOLITE software (release of 26 March 2019) (https://odnature.naturalsciences.be/remsem/software-and-data/acolite (accessed on 22 February 2022)) using the dark spectrum fitting (DSF) atmospheric correction method (e.g., [30–32]) to obtain the multispectral water-leaving reflectance signal ($\rho_w$). The DSF method automatically selects the band producing the lowest atmospheric path reflectance to largely avoid amplification of glint and adjacency effects in the atmospheric correction. This also means that pixels and bands with severe sun glint are discarded from the estimation of the atmospheric contribution and the glint signal is thus still present in the resulting water-leaving reflectance that can drastically affect the derived SPM concentration. The glint reflectance is thus estimated and removed using a glint correction implemented in the ACOLITE software and based on the shortwave infrared (SWIR) band [31,33]. This glint correction results in the decrease in the water-leaving reflectance values by about 0.004 (on average) in the green, red, and NIR bands, which correspond to less than 10% of the total signal in the visible bands and up to 30% in the NIR band for moderate and high turbid waters. For very low turbid waters (<10 g.m$^{-3}$), this effect can nevertheless reach 40% and 100% in the visible and NIR bands, respectively. In addition, a non-water masking (top-of-atmosphere reflectance $\rho t$ at 1600 nm > 0.0215) is applied to remove pixels corresponding to land, above-water objects, clouds, or haze affected by residual glint effects. The final dataset is composed of 56 OLI images and 86 MSI images (Table 1) and provides decent coverage of the period from 2013–2018 and a large SPM concentration range at SORA [0.5–1389 g.m$^{-3}$].

The MODIS dataset is composed of 4 years of MODIS-A and MODIS-T images (2014, 2015, 2016, and 2018). The years 2014 to 2016 correspond to our sea campaigns while MODIS-A images for the year 2018 were added to increase the dataset in common with OLI and MSI acquisitions. Level 1A MODIS-A and MODIS-T data were downloaded from the oceancolor.gsfc.gov website then processed using SeaDAS (version 7.0) software (seadas.gsfc.nasa.gov, accessed on 3 March 2022) to generate geolocation and Level 1B files. Rrs(555), Rrs(645), and Rrs(859) Level 2 products were then generated using the l2gen function. The MUMM [34] atmospheric correction was applied [12]. However, this correction is only valid for moderately turbid waters and can lead to an underestimation of SPM concentration in highly turbid waters sometimes found in the Rhône River mouth. Based on visual inspection, images corresponding to highly turbid waters close to the river mouth were reprocessed using the SWIR atmospheric correction [35]. l2gen flags were used to mask clouds (reflectance threshold at 2130 nm > 0.018) and glint. As MODIS images were used to estimate the Grand Rhône River plume surface, images with less than 80% of valid pixels on the region of plume presence (ROPP) were discarded from the dataset (see [15]). The ROPP was identified using all available images, corresponding to pixels where at least 5% of the SPM concentrations were above 3 g.m$^{-3}$ [15] (Figure A1A) and with a longitude higher than 4.5°E (to focus on the Grand Rhône turbid plume only). In addition, reflectance products showing abnormally high values in the red band all over the study area, likely due to unmasked glint, were also removed from the dataset. Therefore, for each year, the mean water reflectance value in the red band was computed for each image in an offshore SPM free rectangle of the study area (Figure A1A). These "offshore reflectances" were plotted as a function of days and fitted with a Gaussian function. Images for which the difference between the offshore reflectances and fitted reflectances was larger than 1.5 times the standard deviation of the offshore reflectances were considered outliers (Figure A1B). The final dataset is composed of 916 MODIS images, covering 580 days over 4 years. The dataset is well-distributed over seasons, with about 19% of the dataset acquired in winter, 31% in summer and about 25% in spring and autumn (Figure A2A). The distribution of this dataset as a function of the freshwater discharge in Arles is also highly representative, as it is identical to those observed in the 2005–2020 period (61% for dry conditions (Q < 1500 m$^3$.s$^{-1}$) and 39% for wet conditions (Q > 1500 m$^3$.s$^{-1}$)) (Figure A2B).

Only spectral bands useful for SPM retrieval (see Section 2.3) were considered, i.e., the green, red, and near-infrared (NIR) bands at 561/560/555 nm, 655/665/645 nm, and 865/865/859 nm, respectively, for the OLI, MSI, and MODIS sensors, after applying atmospheric corrections (using NIR and SWIR bands). The MODIS red and NIR bands are provided with a spatial resolution of 250 m. The green band has a native lower resolution of 500 m but, for our purposes, it was recomputed at a 250 m resolution for consistency with red and NIR bands. Similarly, the 865 nm NIR band of MSI is provided with a 20 m resolution and was recomputed at a 10 m resolution to be consistent with the MSI green and red bands resolution.

The three sensors' specifications and datasets are summarized in Table 1.

**Table 1.** In situ and satellite datasets used in this study.

| | Satellite/Sensors | Used Spectral Bands (nm) | Spatial Resolution (m) | Temporal Resolution (Days) | Atmospheric Correction | Number of Images (N) | Temporal Coverage |
|---|---|---|---|---|---|---|---|
| **Satellite Dataset** | Landsat-8/OLI | 561 | 30 | 16 | DSF with glint correction [31,32] | 56 | 2013–2018; 2019–2020 (high flooding events only) |
| | | 655 | 30 | | | | |
| | | 865 | 30 | | | | |
| | Sentinel-2/MSI | 560 | 10 | 2–3 | DSF with glint correction [31,32] | 86 | 2016; 2018; 2019–2020 (high flooding events only) |
| | | 665 | 10 | | | | |
| | | 865 | 20 | | | | |
| | AQUA/MODIS TERRA/MODIS | 555 | 500 | 1 | MUMM [34] or SWIR [35] for highly turbid waters | 1211 | 2014–2016 2018 (AQUA only) |
| | | 645 | 250 | | | | |
| | | 859 | 250 | | | | |
| | **Location** | **Parameter** | **Acquisition Method** | | | | **Temporal Coverage** |
| **In situ Dataset** | SORA station (Arles)–Rhône River | River discharge ($m^3.s^{-1}$) | Autonomous measurements at SORA station: daily averaged measurements and 4 h high frequency measurements during high flooding events (Q > 3000 $m^3.s^{-1}$). | | | | 2005–2020 |
| | SORA station (Arles)–Rhône River | SPM concentration ($g.m^{-3}$) | Autonomous measurements at SORA station: daily sampling and 4 h high frequency sampling during high flooding events (Q > 3000 $m^3.s^{-1}$). | | | | 2005–2020 |
| | SORA station (Arles)–Rhône River | POC concentration ($g.m^{-3}$) | Autonomous measurements at SORA station: daily sampling and 4 h high frequency sampling during high flooding events (Q > 3000 $m^3.s^{-1}$). | | | | 2005–2020 |
| | Rhône River plume | SPM concentration ($g.m^{-3}$) | Sampling during field campaigns. | | | | February 2014 February 2015 February 2016 |
| | Rhône River plume | Above-water reflectance | Measured with TriOS portable sensor simultaneously with SPM sampling during field campaigns. | | | | February 2014 February 2015 February 2016 |

### 2.3. SPM Switching Algorithm

In sediment-laden waters, water-leaving radiance is roughly correlated with the ratio between light backscattering by sediment particles ($b_{bp}$ coefficient) and absorption by water ($a_w$ coefficient). As $a_w$ increases and $b_{bp}$ decreases with longer wavelengths, the reflectance at short visible wavebands is more sensitive to low SPM concentration while that at longer wavebands it is more sensitive to high SPM concentration. In addition, the water-leaving reflectance is expected to progressively saturate as the SPM concentration increases. This saturation first occurs for short visible (blue and green) wavebands at low SPM concentrations (<~20 $g.m^{-3}$), then for the red waveband at moderate concentrations (<~80 $g.m^{-3}$), e.g., [36], and even for the NIR wavebands at higher SPM concentrations, e.g., [37]. Estimating the concentration of SPM over a wide range of turbidity is thus challenging and requires switching between bands to maintain a high sensitivity of $\rho_w$ to SPM concentration variations and avoid saturation. To estimate the SPM concentration along a continuum from the Rhône River (SPM concentration up to 6000 $g.m^{-3}$) to the river plume (SPM concentration often lower than 10 $g.m^{-3}$), we (i) built three semiempirical

relationships between the green, red, and NIR bands of the three satellite sensors and the SPM concentration and (ii) developed an algorithm to automatically switch between these relationships based on the saturation point of the red and green bands. These two steps were briefly described hereafter and more details are available in Appendix A.

### 2.3.1. $\rho_w$ vs. SPM Relationships

The $\rho_w$ vs. SPM relationships used in this switching algorithm are based on the Nechad semiempirical relationship [38]:

$$\text{SPM} = A^\rho \times \frac{\rho_w(\lambda)}{1 - \rho_w(\lambda)/C^\rho} \tag{3}$$

where $\rho_w$ is the water-leaving reflectance, $\gamma \approx 0.216$ is a factor accounting for the air-water transmission, and $A^\rho$ (g.m$^{-3}$) is the coefficient to be calibrated. $C^\rho = \gamma\,C/(1-C)$ is an asymptotic coefficient related to the water-leaving reflectance saturation effect [38], with $C = b_{bp}^*/a_p^*$, $b_{bp}^*$ and $a_p^*$ being the SPM mass-specific particulate backscattering and absorption coefficients. The $C^\rho$ coefficient value computed from the literature in [38] was used to constrain our green and red band relationships while it was calibrated simultaneously with the $A^\rho$ coefficient for the NIR relationship (see Appendix A). The $A^\rho$ coefficient (and $C^\rho$ for the NIR relationship) and its 95% confidence interval were estimated by fitting the Nechad $\rho_w$ vs. SPM relationships (Equation (3)) to in situ hyperspectral water-leaving reflectances measured in situ in the Rhône River plume and mouth during the three field campaigns (Figure 3, Table 2). The full SPM concentration range (1–60 g.m$^{-3}$) was used to calibrate the red and NIR relationships while only data with $\rho_w$(G) values lower than 0.06 were used to calibrate the green one (and 3 outliers were removed) (Figure 3). This threshold corresponds to the saturation point of the green band, i.e., the value under which the green band is mainly used to compute the SPM concentration (see Section 2.3.2), and this helps to better calibrate the relationships for these low SPM concentrations (<~20 g.m$^{-3}$) (Figure 3A). In addition, to calibrate the NIR relationship coefficients, higher SPM concentration values measured at the SORA station and simultaneous OLI- and MSI-derived $\rho_w$ values averaged in a $3 \times 3$ pixels box centered on the SORA station were used (Figure 3, Table 2, see Appendix A).

**Table 2.** Recalibrated Nechad $\rho_w$ vs. SPM relationship coefficients for the green, red, and NIR bands of the MSI, OLI, and MODIS sensors. The $A^\rho$ coefficient and its 95% confidence bounds as well as the $C^\rho$ coefficient for the NIR relationship were fitted on the data (Figure 3) while the $C^\rho$ coefficients for the green and red relationships were computed from the literature (*in italic* [38]).

| Sensor Bands | Location | Number of Fitted Data (N) | MSI | | | | OLI | | | | MODIS | | | |
|---|---|---|---|---|---|---|---|---|---|---|---|---|---|---|
| | | | $A^\rho$ (g.m$^{-3}$) | 95% | $C^\rho$ | $R^2$ | $A^\rho$ (g.m$^{-3}$) | 95% | $C^\rho$ | $R^2$ | $A^\rho$ (g.m$^{-3}$) | 95% | $C^\rho$ | $R^2$ |
| Green | Rhône River plume | 21 (field campaigns) | 69 | 57;80 | *0.1449* | 0.60 | 76 | 66;86 | *0.1449* | 0.68 | 66 | 55;78 | *0.1449* | 0.56 |
| Red | Rhône River plume | 90 (field campaigns) | 228 | 212;244 | *0.1728* | 0.72 | 208 | 193;222 | *0.1686* | 0.71 | 193 | 179;206 | *0.1641* | 0.70 |
| NIR | Rhône River plume and SORA station | 89 (field campaigns) 38 (MSI vs. SORA) 32 (OLI vs. SORA) | 2738 | 2524;2952 | 0.1838 | 0.98 | 2743 | 2529;2958 | 0.1835 | 0.98 | 2572 | 2372;2773 | 0.1961 | 0.98 |

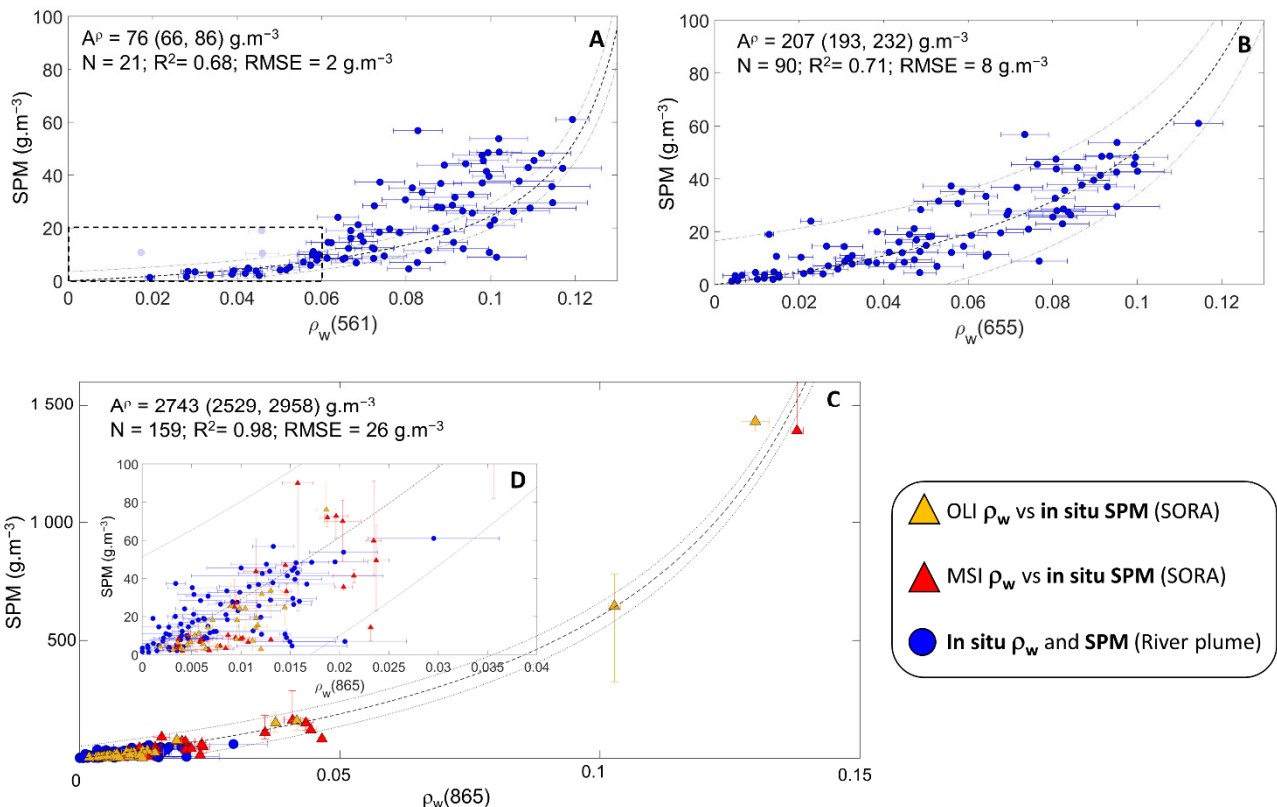

**Figure 3.** Nechad $\rho_w$ vs. SPM relationships obtained for the 555 nm (**A**), 665 nm (**B**), and 865 nm (**C**,**D**) spectral bands of the OLI sensor. (**D**) Zoom on the 865 nm relationship for SPM concentrations lower than 100 g.m$^{-3}$. The fitted Nechad coefficient $A^\rho$ (g.m$^{-3}$) (and $C^\rho$ for the NIR band) as well as its 95% confidence bounds are indicated on the plots and reported in Table 2 with parameter values for other sensors. N is the number of data fitted for each relationship. For the green band, the $A^\rho$ coefficient was calibrated using $\rho_w$ values lower than 0.06 only (dark dotted squares) and three outlier measurements were removed (light blue points). Error bars for in situ data from field campaigns correspond to the standard deviation computed on averaged $\rho_w$ data. Error bars on SPM measured at SORA station correspond to the maximum SPM concentration variations observed at $\pm 1$ day. Error bars on OLI- and MSI-derived $\rho_w$ data correspond to the standard deviations computed within the $3 \times 3$ pixels box centered on the SORA station.

　　　The relationships established for the green, red, and NIR bands are quite similar for the three sensors (Table 2). The calibrated Nechad-derived relationships fit the data well and indicate $\rho_w$ saturations starting with SPM concentrations around 15 g.m$^{-3}$, 40 g.m$^{-3}$, and 400 g.m$^{-3}$, respectively, in the green, red, and NIR bands. The green and red bands fitted for $\rho_w$ vs. SPM relationships show favorable $R^2$ values of 0.68 and 0.72 (Figure 3A,B, Table 2). The NIR $\rho_w$ vs. SPM relationship is well-defined for SPM concentrations larger than 100 g.m$^{-3}$ and up to 1500 g.m$^{-3}$ with a determination coefficient ($R^2$) of 0.98 (Figure 3C, Table 2). For low SPM concentrations (<10 g.m$^{-3}$), data are highly scattered due to the low water-leaving reflectance signal in the NIR. For moderate SPM concentrations (10–100 g.m$^{-3}$), the $\rho_w$(NIR) vs. SPM points follow a linear regime well-described by the calibrated Nechad relationship despite a significant scatter ($R^2 = 0.4$). This scatter can be attributed to measurement uncertainties but also to the high variability of the particle type, composition, and size observed in the Rhône River ([18], Figure 1B), which can affect their optical properties. Nevertheless, RMSE values show that the three bands are able to retrieve the SPM concentration with an uncertainty lower than 13 g.m$^{-3}$ for concentrations between 0 and 100 g.m$^{-3}$ and about 26 g.m$^{-3}$ for higher concentrations.

### 2.3.2. $SPM_{SA}$ Concentration Computation Using the Switching Algorithm

The switching algorithm developed in this study is based on the algorithm developed and validated in [14] for estuarine sediment-dominated waters. This switching algorithm helps estimate the SPM concentration ($SPM_{SA}$) from $\rho_w$ by selecting and automatically switching between the most sensitive green, red, and/or NIR radiometric band(s). $SPM_{SA}$ is thus computed using a combination of the $SPM_G$, $SPM_R$, and $SPM_{NIR}$ values obtained using the $\rho_w$ vs. SPM relationships developed in Section 2.3.1 (Equation (3)). For this purpose, the saturation points of the green ($S_G$) and red ($S_R$) bands were estimated using the following band-to-band relationships: $\rho_w(R)$ vs. $\rho_w(G)$ and $\rho_w(NIR)$ vs. $\rho_w(R)$, respectively (Figure 4). To be the most representative of the $\rho_w$ range encountered over the study area, we built these relationships using all in situ and satellite-derived data available (Figure 4, Appendix A). To avoid too sharp transitions between the use of the SPM concentrations computed from the three bands, the switches are not applied right at the saturation points but progressively through weighting factors ($\alpha$, $\beta$, and $\gamma$, Table 3) within the saturation intervals defined around the saturation points (see Appendix A and Figure A3 for more details on saturation points and radiometric bounds computation). The radiometric bounds of these saturation intervals (G2R and R around $S_G$ and R2N and N around $S_R$, Table 3) are estimated in the red band, which is common to the two band-to-band relationships (Figure 4). The $SPM_{SA}$ concentration is then computed as:

$$SPM_{SA} = \alpha \times SPM_G + \beta \times SPM_R + \gamma \times SPM_{NIR} \tag{4}$$

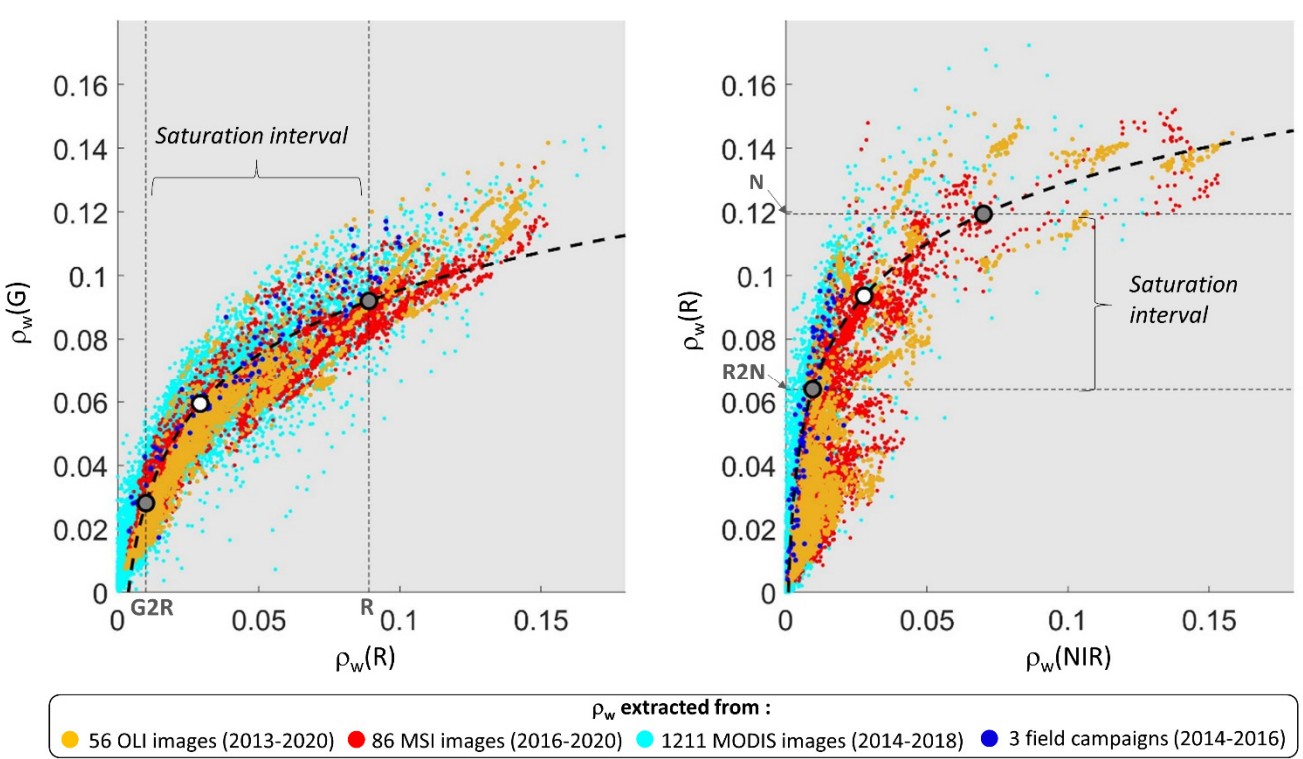

$\rho_w$ **extracted from :**

● 56 OLI images (2013-2020)   ● 86 MSI images (2016-2020)   ● 1211 MODIS images (2014-2018)   ● 3 field campaigns (2014-2016)

**Figure 4.** Switching algorithm saturation points (white circles) and saturation intervals radiometric bounds (G2R, R, N, N2R) (grey points) computed from band-to-band relationships (values for the three sensors are available in Table 3). Satellite-derived $\rho_w$ data were extracted from a transect going from the SORA station to the offshore limit of the river plume for each satellite image (Figure A5, Appendix A). Before computation, the band-to-band relationships were modeled using a logarithmic curve (dashed line) (See Appendix A for more details).

**Table 3.** Saturation intervals and weighting factors ($\alpha$, $\beta$, and $\gamma$) used to compute the $SPM_{SA}$ concentration using the switching algorithm. Saturation interval radiometric bounds values are indicated for each sensor (the green ($S_G$) and red ($S_R$) saturation point values are indicated but not used in the algorithm and the R radiometric bound is not used in the algorithm, see Appendix A).

| $\rho_w(R)$ Interval | Used SPM Concentration | Weighted Factors | | |
|---|---|---|---|---|
| <G2R | $SPM_G$ | $\alpha = 1$ | $\beta = 1$ | $\gamma = 0$ |
| [G2R; R2N] | Saturation interval: $SPM_G$ and $SPM_R$ | $\alpha = \ln\left(\frac{R2N}{\rho_w(R)}\right) \div \ln\left(\frac{R2N}{G2R}\right)$ | $\beta = \ln\left(\frac{\rho_w(R)}{G2R}\right) \div \ln\left(\frac{R2N}{G2R}\right)$ | $\gamma = 0$ |
| [R2N; N] | Saturation interval: $SPM_R$ and $SPM_{NIR}$ | $\alpha = 0$ | $\beta = \ln\left(\frac{N}{\rho_w(R)}\right) \div \ln\left(\frac{N}{R2N}\right)$ | $\gamma = \ln\left(\frac{\rho_w(R)}{R2N}\right) \div \ln\left(\frac{N}{R2N}\right)$ |
| >N | $SPM_{NIR}$ | $\alpha = 0$ | $\beta = 0$ | $\gamma = 1$ |

| $\rho_\omega(R)$ / Sensors | G2R | SG | R2N | SR | N |
|---|---|---|---|---|---|
| MSI | 0.0103 | 0.0301 | 0.0588 | 0.0879 | 0.11 |
| OLI | 0.0102 | 0.0297 | 0.0622 | 0.0924 | 0.1145 |
| MODIS | 0.0102 | 0.0298 | 0.0624 | 0.0936 | 0.117 |

Values of the weighting factors ($\alpha$, $\beta$, and $\gamma$) and radiometric bounds for each satellite sensor are presented in Table 3.

We can notice that the band-to-band relationships presented in Figure 4 emphasize a clear saturation of $\rho_w(G)$ and $\rho_w(R)$ for increasing SPM concentration. The variations of these saturation points (~0.06–0.1 for $\rho_w(G)$ and 0.09–0.14 for $\rho_w(R)$) across the different images probably result from the variability of the SPM mass-specific optical properties reported in the Rhône River plume by [39] and are likely related to variations of SPM size and composition ([40], in agreement with [18] and as highlighted by the POC and SPM variations in Figure 2B). However, the $\rho_w(NIR)$ vs. $\rho_w(R)$ relationship shows an unexpected high scatter at low $\rho_w(R)$ values ($\rho_w(R) < 0.09$) caused by an abnormal increase in NIR reflectance values in the river. These abnormally high NIR reflectance values mainly appear on images recorded during the spring and summer seasons and are likely caused by adjacency effects from surrounding vegetation and/or residual glint (Figure A4). Fortunately, the SPM concentrations are mainly estimated using the red and green bands as the water is moderately turbid during these seasons (Figure A4), and these bands are only weakly affected by adjacency effects, notably based on computations made using the SIMilarity Environment Correction (SIMEC) code [41]. Results obtained show that adjacency effects on the water reflectance can be as low as 5% at 560 and 665 nm and 10% at 865 nm (for the most favorable case of turbid waters and low reflective land, i.e., winter period in our area) and as high as 20% at 560 and 665 nm and 80% at 865 nm for the worst case (clear waters and highly reflective land, e.g., summer period). Therefore, we assume the adjacency effects on satellite-derived water reflectance values and on SPM concentrations to be moderate along the river (up to about 20%), including at the SORA station where the $\rho_w(NIR)$ are consistent with the $\rho_w(G)$ and $\rho_w(R)$ values (Figure A4).

*2.4. Calculation of the River Plume Surface and SPM Mass*

To extract the Grand Rhône River plume surface and SPM mass within the plume, we first developed a routine, adapted from [15], to automatically detect the plume boundaries on MODIS images.

The turbid plume boundaries are detected using the SPM concentration estimated using the red band only ($SPM_R$) as it is the most adapted band to detect the moderate SPM concentrations in the river plume and is less sensitive to the lowest SPM concentrations which tends to blur the plume boundaries. However, a visual inspection of the MODIS images showed that the overall level of the MODIS-derived water-leaving reflectance

values in the red band vary from one image to another. This affects the estimated $SPM_R$ concentration and can complexify the application of a common threshold to delineate the turbid plume. To estimate these variations, for each image, we computed a water-leaving reflectance background value in the red band ($\rho_{w\_background}(R)$) in an offshore SPM free rectangle located south of the turbid plume (latitude < 42.5°N, Figure A1), where the water reflectance at 645 nm is supposed to be negligible. This water-leaving reflectance background follows an annual Gaussian shape with values near 0 in winter and increasing up to 0.01 in summer (Figure A1), which can induce a $SPM_R$ concentration variation of $2$ g.m$^{-3}$. These seasonal variations suggest an effect of the illumination conditions through the varying position of the sun with respect to the sensor. The resulting $SPM_R$ concentration variation of $2$ g.m$^{-3}$ is under our $SPM_R$ concentration estimation uncertainties (Figure 3) but is sufficient to significantly affect the identification of the turbid plume boundaries that are characterized by low SPM concentration and a strong gradient. Therefore, to select and apply a common threshold for all images, the $\rho_w(R)$ values of each MODIS image were corrected from the estimated $\rho_{w\_background}(R)$ following Equation (5), before being converted into an $SPM_R$ concentration.

$$\rho_{w\_Corr}(R) \; = \; \rho_w(R) - \rho_{w\_background}(R) \tag{5}$$

In addition, because of the shallow depth of the Rhône prodelta zone and its exposure to waves, sediment resuspension generates surface SPM that are not directly related to river outputs. As proposed by [15], we therefore removed image pixels with water depths lower than 20 m. This step is moreover useful to better separate the Grand Rhône River plume from the Petit Rhône River plume (see after).

The threshold used to delimitate the turbid plume was defined using the same method as [15]. This method is based on an analysis of the percentile 95 of the corrected $SPM_R$ concentrations in embedded areas centered on the Rhône River mouth, and for all MODIS images. The same trend as [15] is observed with our data, with percentile 95 values slightly increasing from 1.5 g.m$^{-3}$ to a plateau around 2.5–3 g.m$^{-3}$ for large areas, and then suddenly increasing towards the Rhône River mouth (Figure A6). As in [15], we considered that this $SPM_R = 3$ g.m$^{-3}$ concentration plateau corresponds to the limit between the turbid plume and the ambient SPM concentration observed outside, and was thus selected as the threshold. To estimate the sensitivity of the metrics (i.e., plume surface and mass) to this threshold, we also computed the plume boundaries for $SPM_R$ threshold values of $2$ g.m$^{-3}$ and $4$ g.m$^{-3}$ and used them to estimate uncertainties on the turbid plume surface and SPM mass (Figure 5).

Once the threshold was applied to the dataset, we developed a routine to (1) identify the plume contour that corresponds to the Grand Rhône River plume, (2) extract its surface area, and (3) compute the corresponding plume surface mass.

1. The identification of the Grand Rhône River plume was completed by selecting the contour with the minimal distance to the river mouth (here defined as the MesuRho platform pixel) lower than 1 km (Figure 5B). When both the Petit Rhône and Grand Rhône River turbid plumes were merged (mainly under southeastern winds), a routine was used to allow the contour to slightly contract (between 3 and 4 g.m$^{-3}$), using the Chan–Vese active boundaries model [42] in order to separate them as best as possible.

2. The plume surface was estimated by summing the number of pixels within its boundary and was converted to area units (in km$^2$) considering the MODIS spatial resolution of 0.25 km.

3. The SPM mass within the river plume was estimated assuming a 1 m thickness with a homogeneous SPM concentration. This choice of 1 m thickness was mainly based on the optical depth viewed by the satellite sensor in the red spectral band, e.g., in [11]. The $SPM_{SA}$ concentration of each pixel inside the defined turbid plume boundaries were thus multiplied by pixel volume (area $\times$ 1 m) and then summed. Nevertheless, measurements show that the Rhône River surface turbid plume has a thickness varying

between 1 and 5 m depending on the wind direction and distance from the coast, and with a sharp decrease in the SPM concentration within the first meters (e.g., [11,23]). The SPM mass computed in this study thus has to be considered a first approximation of the mass of sediment trapped in the surface plume layer.

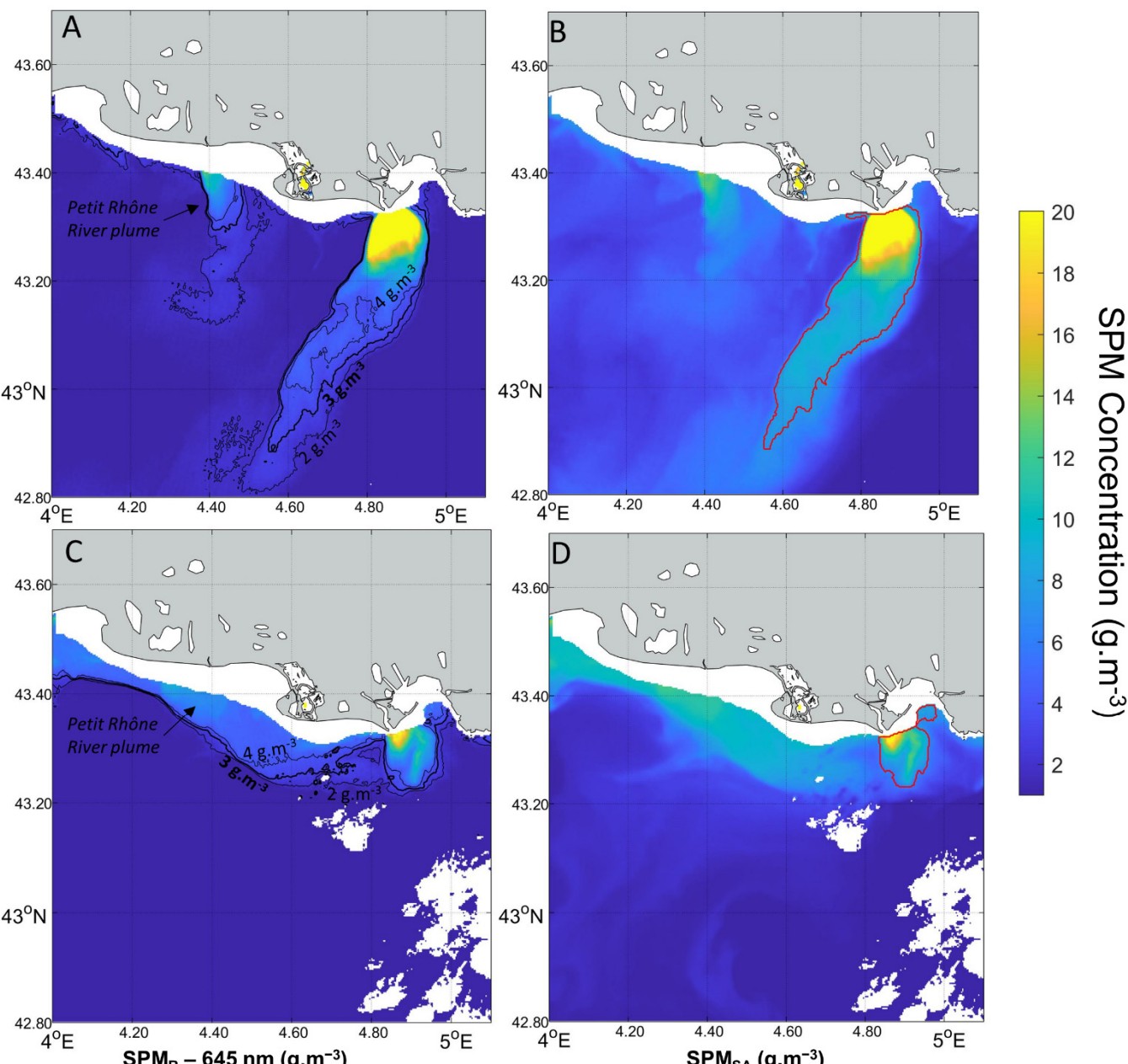

**Figure 5.** Detection of the Rhône River plume boundaries for two wind conditions: offshore wind on 20 February 2014 (**A**,**B**) and onshore wind on 3 March 2018 (**C**,**D**). (**A**,**B**) MODIS-derived $SPM_R$ concentration map with contours corresponding to $SPM_R$ concentrations of 2, 3, and 4 g.m$^{-3}$. The threshold of 3 g.m$^{-3}$ (coarse line) is selected to delimitate the plume contour while the thresholds of 2 and 4 g.m$^{-3}$ (thin lines) are used to evaluate the uncertainties on the derived Rhône River plume area and SPM mass due to the choice of the threshold. (**B**,**C**) Final plume contour (3 g.m$^{-3}$) superimposed on the $SPM_{SA}$ concentration map. For (**D**), the final contour was allowed to slightly contract toward the threshold of 4 g.m$^{-3}$ in order to better isolate the Grand Rhône River plume from the Petit Rhône River plume. White areas along the coast correspond to shallow waters with depths lower than 20 m.

## 3. Results

### *3.1. Validation of the Switching Algorithm*

3.1.1. Illustration of the Switching Algorithm Application

Figure 6 illustrates the application of the switching algorithm to the OLI image recorded on 6 November 2014. It highlights the advantages of using this type of algorithm compared to a single band algorithm. OLI-derived SPM concentration maps presented in Figure 6A–C as well as SPM concentration transects presented in Figure 6H show that $\rho_w$ in any single band cannot accurately estimate the SPM concentration from the SORA station in Arles to the offshore limit of the plume. $\rho_w$ in the green and red bands efficiently estimated the SPM concentration in the very low (<15 g.m$^{-3}$) and moderate turbid waters (<50 g.m$^{-3}$), respectively, but both show a clear saturation for higher SPM concentrations. The NIR band is the only one that aids in detecting the sharp increase and decrease in the SPM concentration at the river mouth and retrieving the high SPM concentrations measured in the river at the SORA station. However, the SPM$_{NIR}$ concentrations were noisy and clearly overestimated in clear to low turbid waters (e.g., SPM$_{NIR}$ concentration values estimated at ~20 g.m$^{-3}$ outside the plume), likely because of a low $\rho_w$(NIR) signal-to-noise ratio, the poorly constrained $\rho_w$(NIR) vs. SPM relationships for these low turbidity waters, and/or a residual glint and adjacency effects (Figure 6H). Figure 6D–F and H show that the switching bounds and weighting factors selected in Section 2.3.2 (and Appendix A) efficiently select the most appropriate SPM concentration range for each band and allow a smooth transition between them. The green band was mainly used to estimate the SPM$_{SA}$ concentration in clear waters and at the very low turbid boundaries of the plume. The red band took over when the SPM$_G$ started to saturate (~10 g.m$^{-3}$) and was primarily used to estimate the SPM$_{SA}$ concentration from the plume to the river mouth. The transition with the NIR band was mainly complete just before the river mouth between 20 g.m$^{-3}$ and 60 g.m$^{-3}$, even though the red band was still used until a high concentration of ~600 g.m$^{-3}$ was reached. This transition helps to avoid using the highly overestimated SPM$_{NIR}$ concentrations and to switch before the complete saturation of the red band. The resulting SPM$_{SA}$ concentration map and transect clearly highlight the plume boundaries and the SPM concentration variations in the plume and in the river (Figure 6G,H).

3.1.2. Matchups Validation

In order to quantitatively validate the switching algorithm and the $\rho_w$ vs. SPM relationships developed in Section 2.3 for retrieving SPM concentration in the Rhône River, 70 matchups were established between the OLI- and MSI-derived SPM$_{SA}$ concentrations at the SORA station (averaged into a $3 \times 3$ pixels box) and the corresponding in situ daily averaged SPM concentration measured at the SORA station (Figure 7A,C). Only data with SPM values at the SORA station presenting uncertainties lower than 100%, i.e., presenting a low to moderate $\pm 1$ day SPM concentration variation were selected. To evaluate the quality of these matchups and the efficiency of the switching algorithm to estimate SPM concentration compared to using a single-band algorithm, root-mean-square errors (RMSE) were computed in low (<10 g.m$^{-3}$), moderate (10–60 g.m$^{-3}$), and high SPM concentration (>60 g.m$^{-3}$) ranges between the in situ data measured at the SORA station and the SPM concentration estimated (i) with each band independently (SPM$_G$, SPM$_R$, and SPM$_{NIR}$) and (ii) using the switching algorithm (SPM$_{SA}$) (Figure 7C). The comparison shows that the switching algorithm efficiently selects the most appropriate band (lower RMSE) to compute SPM$_{SA}$ concentration in each SPM concentration range. Indeed, Figure 7A,C show that for low SPM concentration (<10 g.m$^{-3}$), the green and red bands are used, and both show a very low RMSE of ~2 g.m$^{-3}$ compared to the high RMSE obtained with the NIR band (16 g.m$^{-3}$) because of clearly overestimated SPM$_{NIR}$ values. The red band was then mainly used in moderate turbid waters (10–60 g.m$^{-3}$), where it produced the best RMSE of 14 g.m$^{-3}$, against 21 and 17 g.m$^{-3}$ for the green and the NIR bands, respectively. Finally, the NIR band was used to estimate the high SPM concentration (>60 g.m$^{-3}$) after both the red and the green bands saturated and produced an RMSE of 71 g.m$^{-3}$ against an RMSE

larger than 400 g.m$^{-3}$ for the green and red bands. This effective band selection and switch helps to obtain a final RMSE for the SPM$_{SA}$ concentrations of 2, 13, and 77 g.m$^{-3}$ for low, moderate, and high turbid waters, respectively, and a global RMSE of 37 g.m$^{-3}$ over the whole interval (0–1600 g.m$^{-3}$). This RMSE corresponds to a mean residual of 35% and to a slight underestimation of 4% (linear regression slope of 0.96). The spatial variability of SPM concentrations in turbid and dynamic waters can be an issue when comparing satellite-derived values with field measurements on a fixed point, even for high (meter scale, e.g., [43]) to very high (centimeter scale, e.g., [44])-resolution sensors. The impact of the SPM concentration spatial variability was thus estimated on these matchups by converting the $\rho_w$ standard deviation computed in the 3 × 3 satellite pixels box around the SORA station into an SPM concentration standard deviation (error bars in Figure 7A, which actually do not appear to be lower than the marker size). This impact is limited with a spatial variability lower than ± 15%, which is consistent with [43], where it was observed a spatial variability of water turbidity lower than 10% when downscaling from to 2 to 20 m Pleiades satellite data recorded over highly turbid estuarine waters. This impact is also minimized by the number of field measurements used to compute the daily averaged SPM concentration at the SORA station (16 to 32 measurements a day depending on flood conditions [24]), which smooths the turbidity small-scale variability. The ± 1 day SPM concentration variability at the SORA station is higher, with some data points showing variability larger than 50%. However, removing these data points only slightly improved the mean residual error (33 g.m$^{-3}$), also suggesting a limited impact.

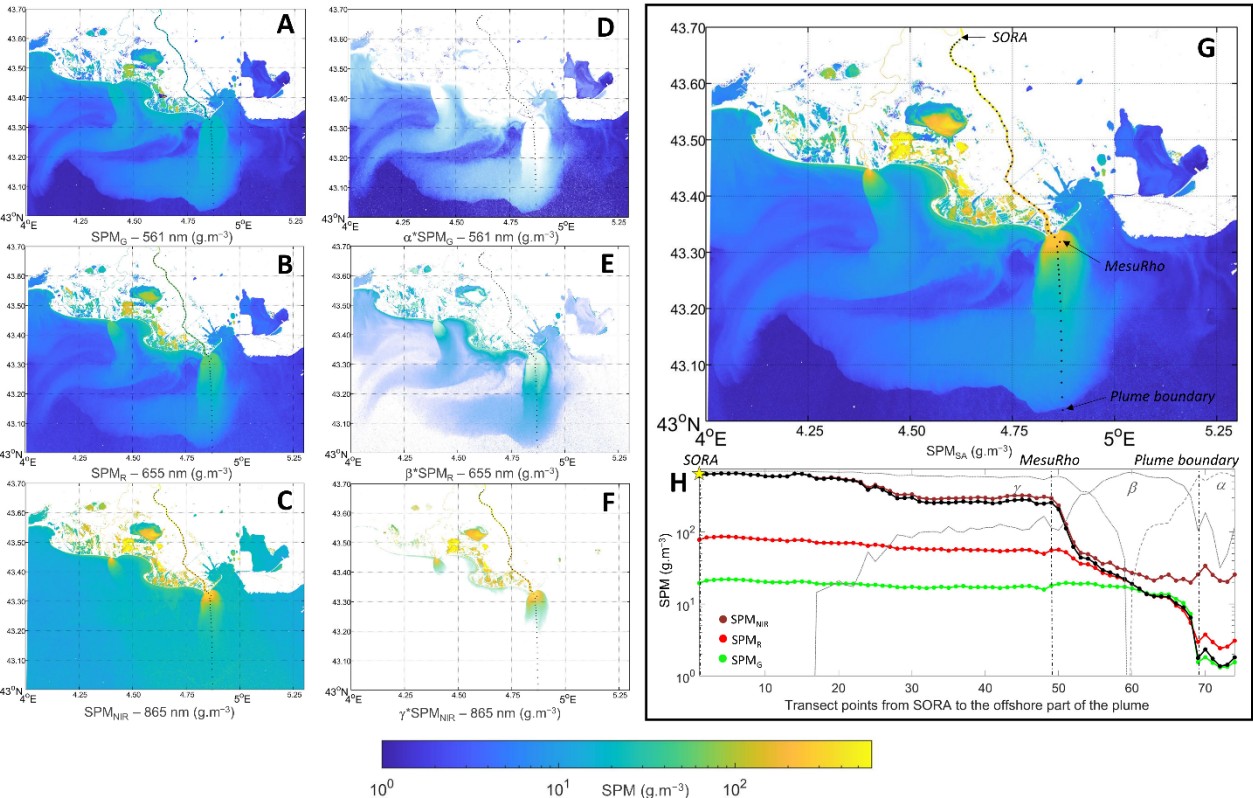

**Figure 6.** OLI-derived SPM concentration maps of the Rhône River and river plume (6 November 2014). The SPM concentrations were estimated using the Nechad $\rho_w$ vs. SPM relationships, recalibrated for the (**A**) green (561 nm), (**B**) red (655 nm), and (**C**) NIR (865 nm) bands, and (**G**) using the switching algorithm (SA). (**D**–**F**) are the same maps as (**A**–**C**) but with their opacity weighted by the α, β, and γ factors: white parts represent regions where the weighting factors equal 0 (i.e., the corresponding SPM$_G$, SPM$_R$, or SPM$_{NIR}$ concentration was not used for the computation of the final SPM$_{SA}$ concentration in these areas) while fully opaque parts correspond to a weighting factor equal

to 1 (i.e., the final SPM$_{SA}$ concentration was computed from the corresponding SPM$_G$, SPM$_R$, or SPM$_{NIR}$ concentration only). SPM concentrations extracted along a transect from the SORA station in Arles to the offshore part of the plume are presented in (**H**) with the SPM$_G$, SPM$_R$, and SPM$_{NIR}$ concentrations represented in green, red, and brown colors, respectively, and the SPM$_{SA}$ in black. The transects point locations are indicated on each SPM map. The values of the weighting factors α, β, and γ along this transect are also reported in grey in (**H**) (values from 0 to 1 were scaled to the maximum SPM concentration for a better visualization). The yellow star corresponds to the in situ SPM concentration value measured at the SORA station on 6 November 2014.

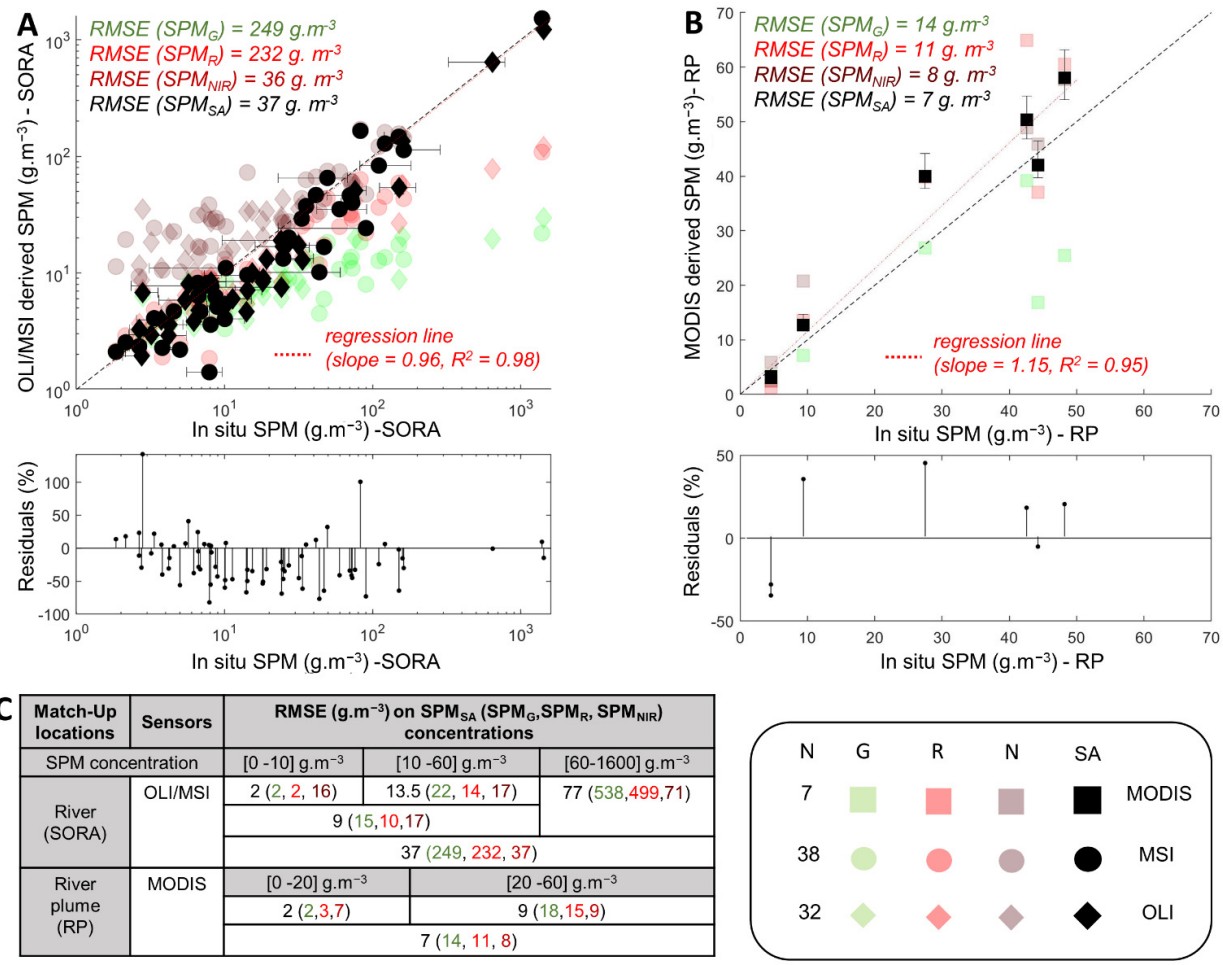

**Figure 7.** Matchups between in situ and satellite-derived SPM concentrations. (**A**) Matchups between the SPM concentration estimated from OLI (number of data *n* = 32, black diamonds) and MSI (*n* = 38, black points) using the switching algorithm (SA) and the in situ SPM concentration measured at the SORA station. (**B**) Matchups obtained between the SPM concentration estimated from MODIS (*n* = 7, black squares) using the switching algorithm and the in situ SPM concentration measured in the river plume (RP) during field campaigns. Error bars on in situ SPM concentration measured at SORA correspond to the maximum SPM concentration interval observed within ±1 day. Error bars on satellite-derived SPM concentrations were computed from the ρ$_w$ standard deviation within the 3 × 3 pixels boxes around the SORA pixel (converted into SPM concentration through the switching algorithm). For comparison, the SPM concentrations estimated in each sensor band using the relationships established in Figure 3 are also represented (green, red, and brown markers for the green (G), red (R), and NIR bands respectively). RMSE for the SPM estimated using the switching algorithm (SPM$_{SA}$) and those obtained using single band algorithms (SPM$_G$, SPM$_R$, SPM$_{NIR}$) are indicated in (**A–C**) for various SPM concentration ranges. Relative residuals (=residuals/[in situ SPM] × 100) are also represented in the bottom panels of (**A,B**).

Nevertheless, we observed that the $SPM_{SA}$ concentrations between 10 and 30 $g.m^{-3}$ were systematically underestimated by about 40%. This underestimation results from the low OLI- and MSI-derived $\rho_w$(R/G) values measured at the SORA station pixel and used to compute the SPM concentration in this 10–30 $g.m^{-3}$ range. For an SPM concentration of ~30 $g.m^{-3}$, these OLI- and MSI-derived $\rho_w$(R/G) values are 20 to 30% lower than the in situ values measured in the river plume and used to calibrate the $\rho_w$(R) vs. SPM relationship (Figure A7). An overcorrection of the red and green bands by the atmospheric or glint correction algorithm could partly explain these lower $\rho_w$(G/R) values measured by the satellite [31]. Hence, the differences observed between the $\rho_w$ values derived from satellite data at the SORA station and those measured in situ in the river plume significantly diminished, especially for $\rho_w$(G/R) in the range 0.04–0.08 (10–50 $g.m^{-3}$), when the glint correction was not activated during image processing (Figure A7). However, it led to abnormally high $\rho_w$(G/R) values (compared to in situ data) at low concentration (<10 $g.m^{-3}$) and abnormally high $\rho_w$(NIR) values for SPM lower than 100 $g.m^{-3}$, due to significant glint contamination (Figure A7). In addition, for high concentrations values (>60 $g.m^{-3}$), the $\rho_w$(G/R) derived from satellite data at the SORA station are still lower than those predicted by the $\rho_w$(G/R) vs. SPM relationships even without the glint correction, suggesting that the glint correction is not only responsible for this difference. Thus, we decided to keep the glint correction in our processing. Another explanation could be that these low $\rho_w$(G/R) values retrieved from satellite data at the SORA station are related to inherent optical properties of the particulate matter inside the river, which could be different from those in the river plume. This would suggest that the $\rho_w$(G/R) vs. SPM relationships calibrated to in situ data measured in the river plume may not be suitable to estimate the SPM concentration at the SORA station, i.e., in the river.

Because of the cloud cover, only seven matchups were obtained between the MODIS-derived $SPM_{SA}$ concentrations and the in situ SPM concentrations measured in the Rhône River plume during field campaigns. For these matchups, the field measurements with the closest time to the MODIS image acquisition were selected and the exact longitude and latitude recorded on the research vessel for these field measurements were used to select the corresponding pixel on the MODIS images. The MODIS $\rho_w$ values were then estimated by averaging inside a $3 \times 3$ pixels box around the field station. The time difference between field measurements and MODIS image acquisitions was less than 1 h for all matchups. These matchups show that the switching algorithm helps to retrieve an $SPM_{SA}$ concentration with an RMSE of 7 $g.m^{-3}$ over the 0–60 $g.m^{-3}$ range into the river plume, which is better than those obtained with each band independently (RMSE between 8 and 11 $g.m^{-3}$). However, the retrieved $SPM_{SA}$ concentrations tended to be overestimated by about 15% (but this value must be taken with caution as it is based on only 7 matchups). This overestimation can be partly explained by a slight overestimation of the MODIS $\rho_w$(R) (~7%) and $\rho_w$(NIR) (~17%) values compared to in situ water-leaving reflectance values measured during campaigns, as revealed by water-leaving reflectance matchups (Figure A8). The regression slope is driven by the SPM concentration higher than 40 $g.m^{-3}$, mainly computed using the NIR relationship. Here, again, the $\rho_w$(NIR) vs. SPM relationship, which was mainly calibrated using the SPM concentration measured at the SORA station in the river and the corresponding satellite-derived $\rho_w$(NIR), may not be suitable to retrieve the SPM concentration in the river plume because of an imperfect atmospheric (glint) correction of satellite data and/or differences in SPM optical properties in the river and in the river plume. Notice that this assumption is supported by the fact that this overestimation disappeared when we replaced the NIR relationship with a relationship calibrated using in situ data from the river plume only (blue relationship in Figure A7C).

These under- and overestimations of the SPM concentration by the switching algorithm highlighted by these matchups and likely related to atmospheric (glint) processing and/or variations in particulate matter optical properties will be discussed in more detail in Section 4. Nevertheless, we consider here that the $SPM_{SA}$ accuracy obtained using the

relationships and switching algorithm defined in Section 2.3 are sufficient for the purpose of this study.

### 3.1.3. SPM Concentration Time Series Using the Switching Algorithm

Figure 8 presents the $SPM_{SA}$ concentrations estimated at the SORA and MesuRho stations using the three satellite sensors from 2014 to 2018. The SPM concentrations estimated by the three sensors were in appropriate agreement considering the differences in spatial resolution and acquisition time [12]. These concentrations help to better describe the SPM concentration variations and peaks recorded at the SORA station during this period. The concentration at the MesuRho station at the river mouth boundary was almost always lower than the concentration observed at SORA, suggesting either a sediment loss along the downstream part of the river or a rapid dilution and sinking of particles just after the river mouth, before the MesuRho station (2 km from the mouth). It can nevertheless reach values higher than 100 $g.m^{-3}$ (up to ~500 $g.m^{-3}$ in December 2016) during high flood events, that is, much higher than measurements recorded in situ so far.

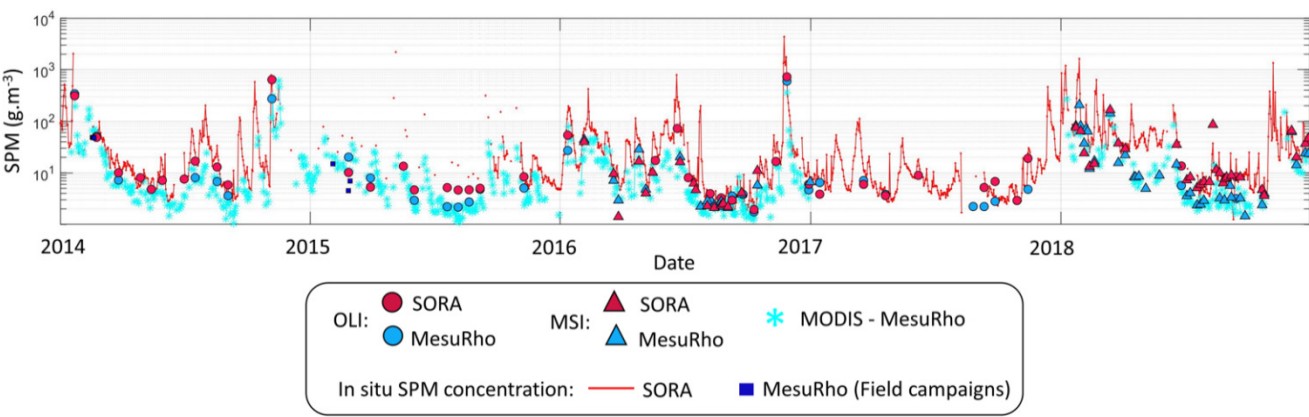

**Figure 8.** Satellite-derived $SPM_{SA}$ concentration at the SORA station (red markers) and MesuRho (blue markers) locations estimated from the three sensors from 2014 to 2018 compared to in situ SPM concentration (SORA: red line; MesuRho (field campaigns): dark blue squares).

### 3.2. Sediment Transport from SORA to MesuRho

For the first time, to the best of our knowledge, we investigated the longitudinal variations of SPM concentrations along the downstream part of a big river (the Rhône), from the gauging station to the first kilometers offshore from the mouth of the river. These transects were exclusively extracted from high spatial resolution satellite data (MSI and OLI data products).

During flood events (SPM concentration > 500 $g.m^{-3}$, corresponding to peak river discharge > ~3000 $m^3.s^{-1}$), various patterns can be observed. Mainly, SPM concentrations decreased with sharp chaotic variations from the SORA gauging station to the mouth, and abruptly decreased (by about a factor 100 from about 1000 $g.m^{-3}$ in the river to 10 $g.m^{-3}$ in the river plume zone) after the mouth (Figure 9A). Rarely, inverse variations can be observed with SPM concentrations increasing towards the river mouth. Only considering medium to low river flow conditions (SPM concentration < 500 $g.m^{-3}$, corresponding to peak river discharge < ~3000 $m^3.s^{-1}$, Figure 3B), the SPM concentrations appeared to be almost constant along the transects, representative of an almost conservative SPM transport up to the river mouth. Then again, past the mouth, SPM concentrations rapidly decreased. On average (Figure 9D), the normalized (with SORA SPM concentration as reference) SPM transport within the surface waters of the river was conservative until the river mouth (but with significant variability), then the concentration drastically fell off over about 10 km to end with a smooth decrease until the river plume offshore limit. The concentration decreased on average of about 25% between the river mouth and the MesuRho station (2 km from the mouth) and reached 70% at 10 km from the mouth

(Figure 9D). The concentration drop occurring before MesuRho appeared to be higher for low river discharges (<2000 m³.s⁻¹), with a decrease of about 45%, against only 10% for higher river discharges (Figure 9C). These results therefore highlight a conservative transport of SPM in the river during low to medium river discharge conditions, and a much more complex behavior of SPM during peak flood events.

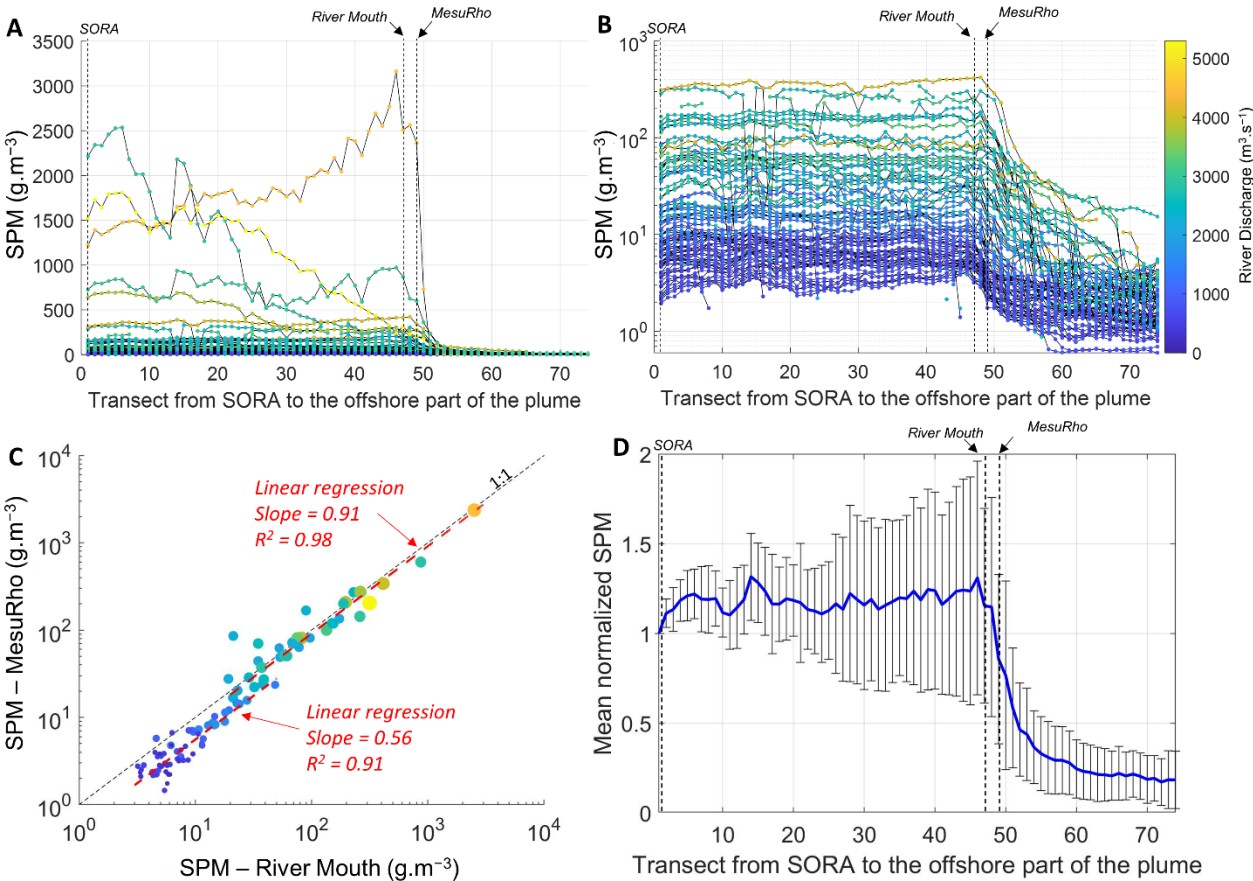

**Figure 9.** SPM concentration variations between the SORA and MesuRho stations, respectively located about 45 km upstream and 2 km offshore from the Rhône River mouth. (**A**) SPM$_{SA}$ concentration transects derived from OLI and MSI satellite data. (**B**) Zoom on the low SPM$_{SA}$ concentration transects (SPM$_{SA}$ concentration at the SORA station < 500 g.m⁻³) with a logarithmic scale. (**C**) Scatterplot between the SPM$_{SA}$ concentrations at the river mouth and MesuRho station extracted from the image transects shown in A. Colors and point sizes depend on river discharge values. Two linear regressions were fitted for low (<2000 m³.s⁻¹) and high (>2000 m³.s⁻¹) river discharges. (**D**) Mean normalized transect. The transects were normalized by the SPM$_{SA}$ concentration at the SORA station, then averaged. The error bars correspond to the standard deviations.

### 3.3. Relationships between the Rhône River Discharge, Plume Area, and SPM Mass

Relationships were established between the daily river water discharge (*x*-axis on Figure 10) and SPM concentrations at SORA and MesuRho, plume area, and plume SPM mass.

The relationship between the SPM concentration and the river liquid discharge is usually represented by a power law [18,45]. To be less sensitive to the high SPM variability during high river discharge conditions, the coefficients of the fitted power law functions were determined using a least squares linear regression between the logarithmic (base 10) values of SPM concentration and liquid river discharge (Figure 10). As could be expected, and despite a high scatter, results highlight two similar relationships at both SORA (in situ data) and MesuRho (MODIS satellite data) stations (Figure 10A). A first relationship

was obtained up to a river discharge of 3000 $m^3.s^{-1}$, with SPM both in the river and at the river mouth increasing in a parallel manner with increasing river discharge. Above this threshold (and up to 5500 $m^3.s^{-1}$), a second sharper relationship was observed, indicating that river waters carry higher loads of SPM per cubic meter, potentially due to enhanced erosion by stronger currents along the draining basin and within the river bed. Two power law relationships must therefore be used to model the increase in SPM concentration in the downstream part of the river with increasing river discharge (Table 4). As already observed (Figure 8), SPM concentrations were most of the time higher at SORA than at MesuRho, but the relationship, as a function of the river discharge, is very similar in the two locations. The fitted power law functions reproduced the relationships well, up to a river discharge of 3000 $m^3.s^{-1}$ ($R^2$ of 0.65–0.70); this was no longer the case during peak flood conditions ($R^2$ values from 0.2 to 0.4), which may originate from different types of flood events (intensity and duration in specific rivers) potentially associated with different types (composition, size) of SPM.

**Table 4.** Fitted coefficients for the relationships obtained between the Rhône River liquid and solid discharge and the SPM concentration at SORA and MesuRho, the plume surface, and the SPM mass (Figures 10 and 11). Coefficients for relationships obtained with the surface (SLB; SUB) and mass (MLB; MUB) lower and upper bounds are also indicated in parentheses (obtained with a plume boundaries SPM concentration threshold at 4 and 2 $g.m^{-3}$, respectively).

| Fitted Power Law: $y = A \times x^B$ | | | | | |
|---|---|---|---|---|---|
| X | y | Discharge Interval ($m^3.s^{-1}$) | A | B | $R^2$ |
| River discharge ($m^3.s^{-1}$) | $SPM_S$ ($g.m^{-3}$) | 300–3000 | $1.462 \times 10^{-05}$ | 1.943 | 0.65 |
| River discharge ($m^3.s^{-1}$) | $SPM_S$ ($g.m^{-3}$) | 3000–5500 | $2.236 \times 10^{-14}$ | 4.46 | 0.41 |
| River discharge ($m^3.s^{-1}$) | $SPM_M$ ($g.m^{-3}$) | 500–3000 | $1.795 \times 10^{-05}$ | 1.777 | 0.70 |
| River discharge ($m^3.s^{-1}$) | $SPM_M$ ($g.m^{-3}$) | 3000–4500 | $8.913 \times 10^{-10}$ | 3.079 | 0.20 |
| River discharge ($m^3.s^{-1}$) | Plume mass ($M_{LB}$; $M_{UB}$) (t) | 500–5000 | $1.122 \times 10^{-11}$ ($7.08 \times 10^{-12}$;$1.622 \times 10^{-11}$) | 4.167 (4.189; 4.182) | 0.66 (0.62; 0.68) |
| Solid discharge ($t.day^{-1}$) | Plume mass ($M_{LB}$; $M_{UB}$) (t) | $1$–$1 \times 10^4$ | 0.0022 (0.0017; 0.0047) | 1.407 (1.383; 1.396) | 0.57 (0.50; 0.62) |
| Solid discharge ($t.day^{-1}$) | Plume mass ($M_{LB}$; $M_{UB}$) (t) | $1$–$7 \times 10^5$ | 2.319 (0.977; 7.849) | 0.71 (0.777; 0.613) | 0.4 (0.42; 0.25) |
| Linear regression: $y = A \times x + B$ | | | | | |
| River discharge ($m^3.s^{-1}$) | Plume surface ($S_{LB}$; $S_{UB}$) ($km^2$) | 500–5000 | 0.1824 (0.1366;0.2683) | $-175.7$ ($-131.3$; $-210$) | 0.57 (0.54; 0.40) |
| Solid discharge ($t.day^{-1}$) | Plume mass ($M_{LB}$; $M_{UB}$) (t) | $1$–$7 \times 10^5$ | 0.06 (0.04; 0.09) | - | 0.68 (0.65; 0.69) |

The area of the river plume as a function of the river discharge is well-represented by a single linear relationship ($R^2$ of 0.54) despite, once again, a rather high scatter (Figure 10B). This relationship is very similar to the one previously established by [15] in using MERIS satellite data. It indicated that the river discharge is logically the main environmental factor controlling the turbid plume dynamics in the adjacent coastal sea. The high scatter observed represents the influences of various wind conditions (e.g., Figure 5) and coastal currents (westward geostrophic flow), which also control the shape and extension of the plume (as will be discussed later) [15].

As for the SPM concentration, the relationship between the SPM mass in the surface (0–1 m depth) plume and the river discharge is best fitted with a power law. This SPM mass was estimated based on a very simplified assumption (the surface plume is 1 m thick and the SPM concentration is constant within this surface layer). It varies from 2 tons (minimum river discharge generating a plume that can be detected using MODIS satellite data) to more than 40,000 tons during the maximum peak flood events covered by our dataset.

### 3.4. Transfer of Suspended Particulate Matter from the River to the River Plume

Here, we analyzed the mean relationships established between the solid discharge at the gauging station (SORA) and the estimated mass of SPM within the surface river plume. A specific goal was to evaluate, for the period of one day, the amount or percentage of SPM discharged by the river into the coastal sea that remains in suspension within surface waters (although the river plume mapped from one satellite image usually reflects several consecutive days of river discharge). This should indirectly indicate the amount

of the riverine SPM trapped in the downstream part, settled in the prodelta zone, or transported offshore by intermediate nepheloid layers and close to the bottom through the BNL (see [23]).

A linear regression was first fitted to the relationship (see the associated equation and statistics in Table 4). The scatterplot (Figure 11A) first shows that all points are below the 1:1 line (except very few points on this line). Points are far below the 1:1 line for the minimum solid discharges then move closer as the solid discharge increased. Therefore, the relationship is not really linear over the whole dataset, despite a significant determination coefficient of 0.68. Depending on the definition of the river plume limits, the corresponding slopes of the linear relationships vary from 0.04 to 0.09, which means that, as a rough approximation, less than 10% of the total SPM transported by the river through the SORA gauging station ends up in the surface river plume. This implies that a massive sinking and probably settling of SPM occurred in the prodelta zone, i.e., right after the river mouth (as highlighted in Figure 9), probably resulting from intensive flocculation processes when SPM reached salty waters. The relationship was best fitted using two power laws with a switching point at $10^4$ t.day$^{-1}$, corresponding to a river liquid discharge of about 3000 m$^3$.s$^{-1}$, similar to those observed for the SPM concentration relationship (see associated equation and statistics in Table 4).

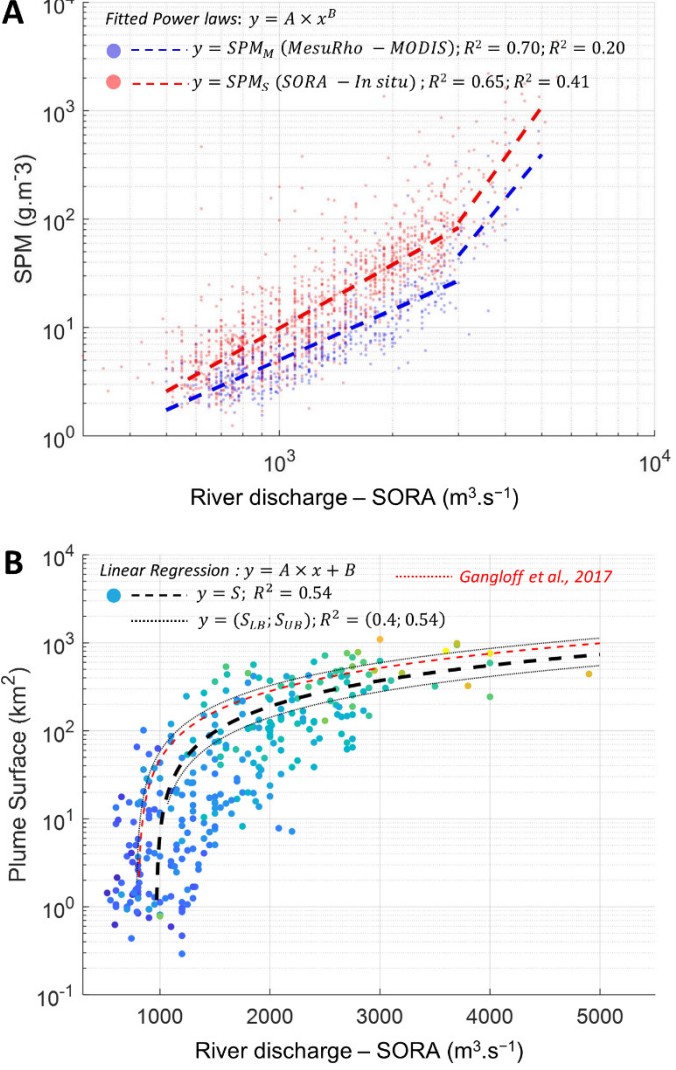

**Figure 10.** *Cont.*

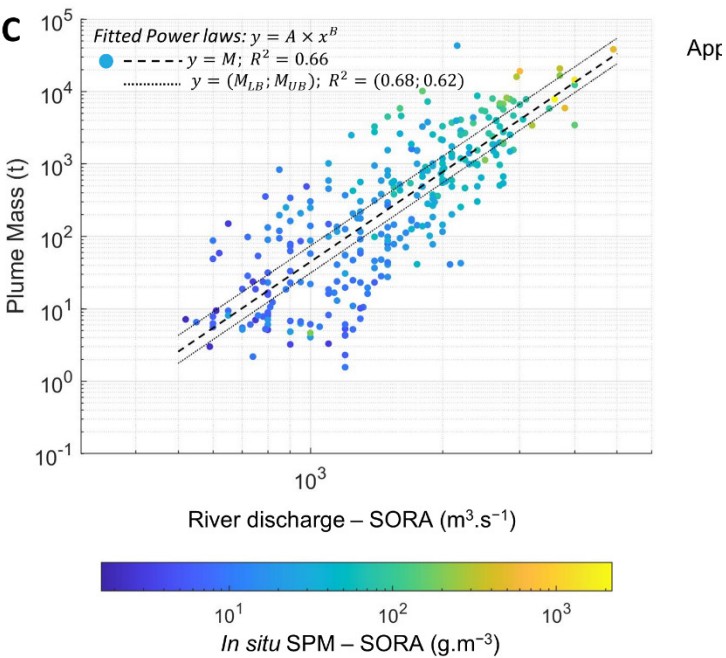

**Figure 10.** Relationships between the Rhône River discharge measured at SORA and (**A**) the SPM concentrations measured at the SORA station (SPM$_S$, red points) and estimated at the MesuRho platform (SPM$_M$, blue points), (**B**) the Rhône River plume surface (S), and (**C**) the Rhône River plume SPM mass (M). The SPM concentration at the MesuRho platform as well as the Rhône River plume surface and SPM mass were estimated using MODIS data (2014, 2015, 2016, 2018). Fitted relationships are represented with dashed lines and coefficients are given in Table 4. Two power laws were fitted to the river discharge vs. SPM concentration relationships, one for river discharges ranging from 500 to 3000 m$^3$.s$^{-1}$ and the other one for river discharges larger than 3000 g.m$^{-3}$. Dotted lines on B and C represent the fitted relationships for plume surface and SPM mass obtained with a plume boundaries threshold of 2 g.m$^{-3}$ and 4 g.m$^{-3}$ and thus represent the plume surface and mass upper (UB) and lower bounds (LB), respectively. All relationship coefficients are given in Table 4.

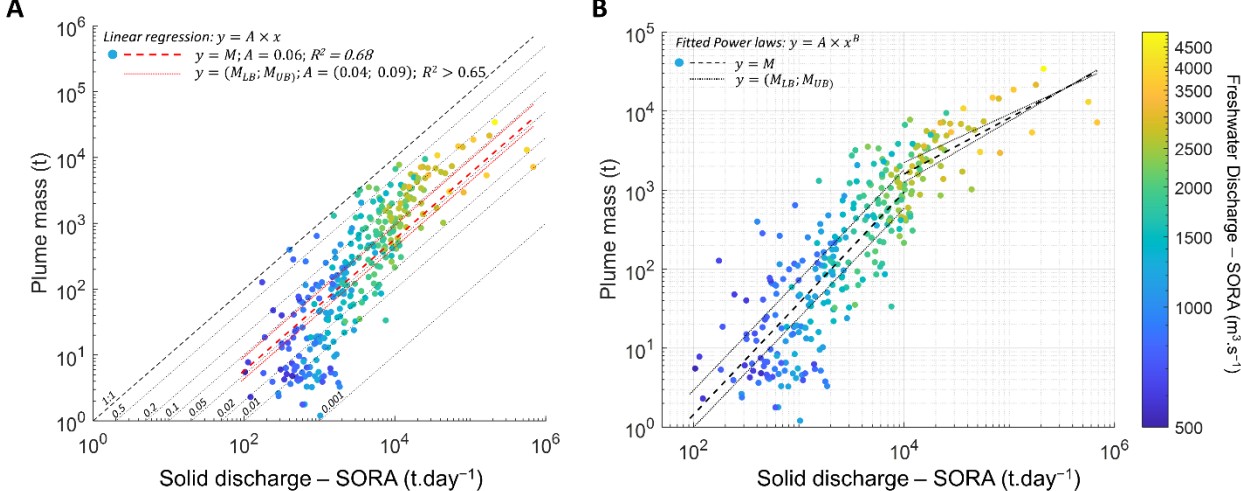

**Figure 11.** Relationship between the Rhône River solid discharge at the SORA station and SPM mass in the Rhône River plume computed using MODIS data (2014; 2015; 2016; 2018). (**A**) Data were first

fitted with a linear regression (dashed line) and the slope was estimated after a logarithmic transformation of data ($log_{10}(y) = A + log_{10}(x)$). For comparison, lines with slope from 0.001 to 1 are also plotted (black dotted lines). (**B**) Data were then fitted with two power laws, one for solid discharges ranging from 100 to $10^4$ t.day$^{-1}$ and the other one for higher river discharges (black dashed lines). For A and B coefficients, dotted lines represent regressions fitted for the plume SPM mass lower (LB) and upper bounds (UB).

## 4. Discussion

### 4.1. The Switching SPM Algorithm

Ref. [14] developed the multi-conditional SPM algorithm, which combines the sensitivity of three wavebands (green, red, and near-infrared) to estimate SPM concentrations over the very large range (1–1000 g.m$^{-3}$) encountered in rivers, estuaries, and river plumes. The original switching methodology between the green, red and NIR spectral bands was established based on field data, i.e., specific to study areas. Here, the method was adapted to a river to river plume system, presenting a similar SPM range, and the switching bounds (Table 3) were directly established based on a high number of satellite data corrected for atmospheric effects and representative of various seasonal conditions. The estimation of the SPM concentration was completed through three calibrated $\rho_w$ vs. SPM concentration relationships. The green and red bands relationships were calibrated using in situ data acquired over the river plume while the NIR relationship was established using satellite-derived water-leaving reflectance and in situ SPM concentration data acquired in the river (SORA station) to calibrate high SPM concentrations. This hybrid set of relationships has the advantage of limiting the impact of potential satellite-derived artifacts on the relationship while allowing the estimation of the high SPM concentration observed in the river.

The main limitations of this switching algorithm come from (1) the estimation of the radiometric effects, mainly glint and/or adjacency effects due to the close proximity of vegetation, and (2) the apparent complexity of the relationship between the water-leaving reflectance and the SPM concentration. If glint and adjacency effects are now well-corrected or well-avoided in open and coastal waters, they are still major issues for the water-leaving reflectance estimation in narrow inland waters such as the Rhône River. The present study shows that the adjacency effect seems to significantly affect the water-leaving reflectance in the NIR band only for low SPM concentration (<60 g.m$^{-3}$) and during spring and summer. This likely results from the growing vegetation in the surrounding cultivated fields during spring that leads to a signal increase in the NIR range because of the well-known vegetation red edge [46] (the increase in $\rho_w$(NIR) in the land surrounding the Rhône River in spring and summer periods was clearly observed on series of MSI images in 2018). These abnormally high values of $\rho_w$(NIR) could also be partly due to residual glint effects as they can significantly affect the water-leaving reflectance, especially in the NIR range. A correction was applied in our study but could be insufficiently to totally correct for this effect in the NIR. This adjacency (and/or residual glint) effect has nevertheless a limited impact on our study as it seems to be minimal at the SORA station located in the city of Arles, possibly because of less surrounding vegetation and because the SPM concentrations lower than 60 g.m$^{-3}$ are mainly estimated using the green and red bands, which seem only weakly affected (<20%) by this effect (Figure A4). Matchups presented in Figure 7 highlight significant differences between the $\rho_w$ derived from satellite data in the river (SORA station) and those measured in situ in the river plume for the same SPM concentration. These differences are clearly visible in Figure A7A,B, which show two distinct $\rho_w$(G/R) vs. SPM concentration relationships for the plume and for the SORA station. The relationship calibrated using data acquired at the SORA station (river) is characterized by higher $A^\rho$ and lower $C^\rho$ coefficients than those calibrated with data from the river plume, resulting in a water reflectance saturation occurring at lower water-leaving reflectance values. The same trends were observed for the NIR relationships, even if high SPM concentration measurements in the river plume were lacking to make a decent comparison (Figure A7C). As already discussed in Section 3.1.2, the difference between these two relationships can be

partly explained by an overcorrection of glint effects by the ACOLITE software. However, this difference is lower but still significant without the glint correction and thus cannot be attributed to an overcorrection of glint effects only. Adjacency effects in the river should lead to an increase in the $\rho_w$ and thus cannot be responsible for this difference. Differences in SPM mass-specific optical properties (absorption and backscattering coefficients per unit of SPM concentration) between the river and the river plume could also explain the difference observed between these relationships. This would suggest variations in SPM size, shape, type, and/or composition between the SORA station and the river plume. Variations could occur at the river mouth and be related to complex flocculation processes in this fresh water to saline water transition zone, in addition to variations in SPM optical properties depending on which main tributaries (Saône, Isère, Durance, and/or Ain Rivers) contribute to the Rhône River discharge [18].

The differences between $\rho_w$ derived from satellite data in the river (SORA station) and those measured in situ in the river plume led to some under- (~40% for SPM concentration between 10 and 30 g.m$^{-3}$) and overestimations (15%) of the SPM concentration at the SORA station and in the river plume, respectively. The overestimation of OLI- and MSI-derived SPM concentrations at the SORA station can impact the SPM concentration variations along the SORA to MesuRho transect presented in Figure 9 but its estimation remains difficult to assess. The slight overestimation (15%) of the SPM concentration in the river plume could lead to an overestimation of the plume mass. This overestimation seems to primarily affect the high SPM concentrations (>~40 g.m$^3$) computed using the NIR relationship, which are mainly encountered in the river and in the river mouth area. This overestimation thus has a very limited impact on the estimated SPM plume mass (< 10%,) and negligible impact on the relationships between the liquid and solid discharges (impact is estimated by replacing the NIR relationship with a relationship calibrated to the in situ data from the river plume (blue relationship in Figure A7C)). The improvement of this switching algorithm will require a better estimation of the adjacency and glint effects as well as the variations of the SPM mass-specific inherent optical properties along the river to river plume continuum. This will necessitate the acquisition of radiometric and sedimentary (SPM concentration, size, shape, and composition) data along this continuum simultaneously with satellite data acquisitions.

Despite the uncertainties discussed before, the current version of the SPM switching algorithm already proved to be satisfactory in retrieving SPM concentrations along rather narrow river transects and was validated based on matchups with daily field measurements carried out at a river gauging station (Figure 7). The SPM concentration spatial variations in the river are lower than 15% and have a very limited impact on high spatial (10 to 30 m) satellite-derived concentrations. Daily averaged measurements of the SPM concentration seem to be sufficient to obtain an appropriate agreement with satellite estimations, except during flooding events when SPM concentration can show a ±1 day variation higher than 100%. The benefits of using a switching algorithm instead of a single-band algorithm was clearly demonstrated over a wide range of SPM concentrations (RMSE of 37 g.m$^{-3}$ from 5 to 1200 g.m$^{-3}$). Such SPM algorithms, switching between different spectral bands, represent an optimal solution in contrasted coastal waters [14,47,48]. The capability to monitor SPM concentrations along rivers using high spatial and now high temporal satellite data could be used in the future to monitor the amount of terrestrial SPM transported by rivers up to the coastal ocean.

*4.2. Transport of SPM from the Downstream Part of the River to the Offshore Limits of the River Plume*

At the Rhône gauging station (SORA), the variations of SPM concentrations as a function of the river freshwater discharge is complex. As a first approximation, SPM concentrations increase exponentially with increasing river discharge, but variations in the relationship are observed depending on the flood event and its intensity (Figure 10A). It may be related to variations in the relative contribution of each tributary to the total

discharge and to the mineral composition of the Rhône. This complex relationship is well-highlighted in Figures 2A and 10A, as well as by the strong variability of SPM composition in the Rhône River represented by the POC-SPM ratio (Figure 2B). These various types of flood events and variability of SPM composition (and most probably size) certainly affect the SPM transport through the river mouth and along the river plume.

The tracking of SPM concentrations along transects from the Rhône River gauging station to the river mouth is another innovative result in our study. Solid fluxes measured at river gauging stations, several tens of kilometers upstream from the river mouth, are usually assumed to be the fluxes discharged to the coastal sea. Here, we actually observed that SPM transport of the downstream part of the river is actually conservative during low to medium river flow conditions (up to ~3000 $m^3.s^{-1}$ in the Rhône River). This is no longer the case during peak flood conditions (>~3000 $m^3.s^{-1}$): most of the time the SPM concentration significantly decreases up to the river mouth, which probably means that a significant number of suspended particles sink and settle down along this section of the river, at least temporarily, and do not reach the sea (Figure 9). In a few cases during high river flow conditions, the SPM concentration was observed to increase towards the river mouth, which probably corresponds to the end of a flood event captured on a satellite image; it may also result from the resuspension of bottom sediments by strong river currents close to the river mouth and/or erosion of sediments from the river sides. Finally, the two robust linear relationships ($R^2 > 0.91$) observed between the SPM concentrations at the river mouth and MesuRho (Figure 9C) indicate that a strong dilution and sinking of SPM occurs after the river mouth. These effects of strong dilution and sinking seem to occur sooner and/or faster for low river discharge (<2000 $m^3.s^{-1}$), with 45% (slope = 0.56) of the SPM concentration already lost at the MesuRho station against only 10% for high river discharge.

Using 10 years of MERIS satellite products, ref. [15] studied the dynamics of the Rhône River plume as a function of meteorological forcings (river discharge and wind conditions). They established the relationship between the plume extent and river discharge, with a very similar slope and determination coefficient, as in the present study ($R^2 = 0.54$), which confirms the predominant role played by the river discharge while wind conditions mainly control the shape and orientation of the plume. Here, we went further by estimating the SPM mass within the river plume (considered 1-m thick homogeneous surface layer) and also logically obtained a significant relationship ($R^2 = 0.66$) between the SPM mass in the river plume and river discharge. It was interesting to relate the river solid discharge (in $t.day^{-1}$) to the SPM mass in the river plume estimated on the same day. The linear relationship obtained ($R^2 = 0.68$) is encouraging in the scope of quantifying the mass of SPM transported offshore and, by difference, the mass of SPM that settles and accumulates in the prodelta zone. Considering the slope of this linear relationship, which varies from 0.04 to 0.09, it clearly appears that only a fraction (6% on average) of SPM discharged by the river is transported offshore within the surface plume. Therefore, more than 90% of the SPM discharged by the river apparently rapidly sinks and settles in the prodelta zone, or is transported in intermediate nepheloid layers. These results are obtained considering a plume thickness of 1 m, but this settling stays over 70% (slope of the linear relationship = 0.28), considering a plume thickness of 5 m. This massive settling of riverine particles will then potentially experience several resuspension and transport processes induced by strong winds and waves along the shallow coastal zone, i.e., only a part of it will finally accumulate in the sediments of the prodelta zone. In the present study, the main objective was to highlight the potential of ocean color satellite data to monitor the transport of SPM along a river to river mouth to river plume continuum. In order to establish a mass balance between the amounts of SPM discharged by the river, settling in the delta zone and transported offshore, a more substantial study is required with a consideration for different time periods (e.g., several days corresponding to a flood event) and a more detailed 3D representation of the river plume (thickness depending on wind direction and distance from the coast) to relate the river solid discharge to the mass of SPM within the river plume.

## 5. Conclusions

Ocean color remote sensing data, now associated with high and very high spatial resolutions, can be used to retrieve SPM concentrations in rivers, particularly from river gauging stations to the river mouths, i.e., along 10 km to 100 km distances where the transport of SPM is not necessarily conservative as assumed in budgets concerning exchanges between land and sea (e.g., [49–52]). The possibility to now use radiometric satellite data for the monitoring of SPM transport along the downstream part of rivers, up to the coastal ocean, was demonstrated in the case of the Rhône River. The data processing is rather complex (due to atmospheric, glint, and adjacency effects as well as variations in composition and size of suspended particles along this continuum), but the results obtained are encouraging. The same method can be applied to any river with similar dimensions, i.e., rivers with sections of several hundred meters. This could better estimate the amount of SPM transported by rivers that actually reaches the coastal ocean and, by difference, the fraction of SPM trapped in the downstream part of rivers (e.g., temporarily trapped in nearshore mudbanks).

While the methodology still requires improvements, our original study has shown that ocean color satellite data at high and medium spatial resolutions can be used to better estimate the mass of SPM transported by rivers that are actually discharged to the coastal ocean and are transported offshore within the surface plume. In order to determine the amount of SPM transported offshore in intermediate or bottom nepheloid layers, the use of autonomous profiling platforms will be required (e.g., [11,23,53]) to complement satellite observations, as well as sediment traps in the delta zone to quantify the settling of SPM within surface sediments [54]. Our preliminary results suggest that only a small fraction (<10%) of SPM discharged by the Rhône River is transported offshore within the surface river plume in the Gulf of Lion. The combined use of satellite observations, autonomous profiling platforms, and sediment traps will be required to constrain numerical models able to reproduce the transport of SPM discharged by rivers in the coastal ocean, determine its fate from the delta zone up to the limits of the continental platform, establish mass budgets, and detect temporal trends associated to climate change and human activities.

**Author Contributions:** A.O. and D.D. conceptualized the manuscript, discussed the methodology, and wrote the original draft. A.O. investigated and made the data analysis. All authors participated in the field campaigns. R.V., F.B., I.P. and A.G. provided their expertise on the Rhône River dynamics. G.P.M. provided his expertise on satellite data processing, hydrology, and geochemistry. All authors participated in the improvement of the manuscript. All authors have read and agreed to the published version of the manuscript.

**Funding:** This research was funded by the CNES-TOSCA (TTC project, P.I. D. Doxaran) and EC2CO (DeltaRhône project, P.I. C. Rabouille) French research programs. Field campaigns were partially funded by the ANR MATUGLI Projet-ANR-14-ASTR-0021.

**Acknowledgments:** Thank you to P. Raimbault and the MOOSE program (Mediterranean Oceanic Observing System for the Environment) now included in the RI ILICO for providing the SORA dataset (permission has been obtained In situ data of SPM properties were acquired through field campaigns supported under national programs TUCPA (EC2CO DRILL 2013) and MATUGLI (ANR-14-ASTR-0021).

**Conflicts of Interest:** The authors declare no conflict of interest.

## Appendix A

*Appendix A.1. $\rho_w$ vs. SPM Concentration Relationships*

To compute the $\rho_w$ vs. SPM concentration relationships for the three sensors, the in situ hyperspectral water-leaving reflectance was first weighted by the spectral sensitivity of the sensors' green, red, and NIR wavebands to obtain the equivalent $\rho_w$ values in the bands of the three sensors (Figure A9).

The OLI- and MSI-derived $\rho_w$ values used for the NIR relationship were extracted from a $3 \times 3$ pixels box centered on the SORA station, defined as the pixel located in the middle of the river (to minimize adjacency effects) in front of the SORA station (same latitude). The standard deviation values of the OLI- and MSI-derived $\rho_w$ values inside the $3 \times 3$ box were reported as an error bar in Figure 3C. To be used in all sensor relationships, the OLI- and MSI-derived $\rho_w$ values were wavelength-shifted to the MSI, OLI, and MODIS bands using factors derived from field campaigns in situ data (Figure A9). In addition, to obtain a robust relationship, only data with SPM values at the SORA station presenting uncertainties lower than 100%, i.e., presenting a low to moderate $\pm 1$ day SPM concentration variation were used.

The Nechad $A^\rho$ coefficients are calibrated by fitting the semiempirical Nechad relationship (Equation (3)) on these in situ and satellite-derived $\rho_w$ vs. SPM concentration relationships using a nonlinear least squares fitting method.

The $C^\rho = \gamma\, C/(1 - C)$ coefficient is an asymptotic coefficient related to the water-leaving reflectance saturation effect [38], with $C = b_{bp}*/a_p*$, $b_{bp}*$, and $a_p*$ being the SPM mass-specific particulate backscattering and absorption coefficients. This coefficient has no impact in the linear regime of the $\rho_w$ vs. SPM concentration relationships. For the green and red relationships, which are mainly used in this linear regime (before the saturation of the bands, see Section 2.3.2. and hereafter), we thus used fixed values computed from "standard" IOP data from the literature in [38]. These "standard" IOP values are in agreement with values observed in the Rhône River plume [39] and allow an appropriate fit of the green and red $\rho_w$ vs. SPM concentration relationships with $R^2$ equal to 0.68 and 0.71, respectively. On the contrary, the high SPM concentration values are mainly estimated using the NIR band relationship in its saturation part, making the $C^\rho$ coefficient a crucial parameter for the SPM concentrations estimation. This coefficient depends on the mass-specific particulate absorption coefficient in the NIR ($a_p*$(NIR)), which is commonly assumed to be null in the "standard" IOP values ($\sim 6.10^{-4}$ $m^2.g^{-1}$ at 770 nm in [38]) while several studies show evidence of significant NIR absorption, especially by mineral particles (e.g., [39,55,56]). In the Rhône River plume, ref. [39] measured a mean mass-specific particulate absorption coefficient of 0.0084 $m^2.g^{-1}$ at 770 nm. This NIR absorption is negligible at low SPM concentration, where the pure water absorption dominates but can become significant in highly turbid waters. To obtain a well-calibrated NIR relationship in both linear and saturation regimes and to take into account a non-null particulate absorption coefficient in the NIR, we thus decided to keep the $C^\rho$ coefficient as a free parameter and to calibrate it simultaneously with the $A^\rho$ coefficients (for the NIR relationship only). The fitted values of $C^\rho = [0.1835–0.1961]$ for the three sensors are lower than those computed from the "standard" IOP values in [38] ($C^\rho = [21.07–21.15]$) but are in agreement with $C^\rho$ coefficient values computed from the mean IOP values measured in the Rhône River plume by [39] and using Equations (A1) and (A2), with $C^\rho$ (865 nm) = 0.2 using $a_p*$(443 nm) = 0.021 $m^2.g^{-1}$ and $b_{bp}*$(532 nm) = 0.011 $m^2.g^{-1}$, and with $C^\rho$ (865 nm) = 0.15 considering mean values directly measured in the NIR of $a_p*$ (770 nm) = 0.0084 $m^{-1}$ and $b_{bp}*$(770) = 0.0094 $m^{-1}$ [39].

$$b_{bp}* = b_{bp}*(532\ nm).*(\lambda/532).\hat{}(-1*\gamma)\ \text{with}\ \gamma = 1 \tag{A1}$$

$$a_p* = a_p*(443\ nm).*\exp(-S_{ap}.*(\lambda - 443))\ \text{with}\ S_{ap} = 0.009\ nm^{-1} \tag{A2}$$

*Appendix A.2. Computation of Switching Algorithm Radiometric Bounds*

The radiometric bounds define the switch from one band to another based on the saturation point of the most sensitive band; thus, on the green saturation point ($S_G$) for the switch from the green to the red band and on the red saturation point ($S_R$) for the switch from the red to the NIR band. The green and red saturation points are estimated using the following band-to-band relationships: $\rho_w$(R) vs. $\rho_w$(G) and $\rho_w$(NIR) vs. $\rho_w$(R), respectively. To be the most representative of the $\rho_w$ range encountered over the study area, we built these relationships using all in situ and satellite-derived data available (Figure A3). For

each OLI and MSI satellite image, $\rho_w$ values were extracted from 74 points along a transect from the "highly turbid" SORA station pixel to the "low turbid" offshore part of the plume (Figure A5). For each transect point, $\rho_w$ values were averaged over a $3 \times 3$ pixels box. This provided $\rho_w$ values over a wide range of turbidity and helped to draw well-defined band-to-band relationships for each image. The same method was applied for each MODIS image but with a transect starting at the river mouth instead of the SORA station, since the MODIS spatial resolution is too coarse inside the Rhône River. In order to compute the saturation points ($S_G$ and $S_R$) for each sensor, the $\rho_w$ values extracted from OLI, MSI, and MODIS data were wavelength-shifted to each sensor band using factors derived from field campaigns in situ data (Figure A9).

Once the band-to-band relationships were obtained for each sensor, they were modelled using a logarithmic regression curve (Figure A3A). This curve starts as linear for the low reflectance values and bends at the point where the saturation of the most sensitive band starts. To allow an appropriate fit of the green and red bands' saturation at a high SPM concentration and to be less sensitive to the $\rho_w$(NIR) values affected by adjacency or residual glint effects that tend to pull down the regression curves, both regressions were weighted by a factor equal to $\rho_w(G)^2$ and $\rho_w(R)^2$, respectively. The actual values of the green and red saturation points ($S_G$ and $S_R$) were then computed as the tangents of the curves equal to 1, i.e., as the middle point between a completely horizontal (complete saturation) and a completely vertical line (Figure A3B).

To avoid too sharp transitions between the use of the SPM concentrations computed from $\rho_w$ in the three bands, the switches are not applied right at the saturation points but progressively through weighting factors ($\alpha$, $\beta$, $\gamma$) inside each saturation interval (Table 3). These saturation intervals delimit the radiometric regions around the two saturation points where the band-to-band relationships are no longer linear. Therefore, for the two band-to-band relationships, points of the logarithmic regression curves located before and after the saturation points were approximated with a linear regression. Saturation interval radiometric bounds (G2R and R around $S_G$ and R2N and N around $S_R$) are then considered to be the points where the tangent of the logarithmic regression curves equals these linear approximations (Figure A3B,C). These radiometric bounds are estimated in the red band, which is common to the two band-to-band relationships (Figure A3C, Table 3). We can see that the upper bound of the green band saturation interval (R) shows a higher reflectance value than the lower bound of the red band one (R2N). We thus decided to only use the R2N bound as a switching point (Table 3).

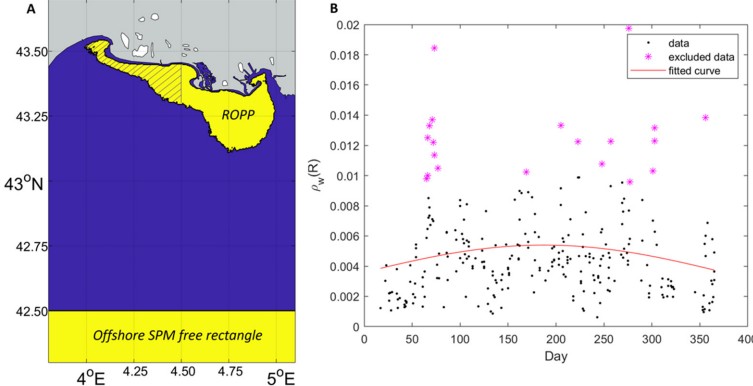

**Figure A1.** (**A**) Delimitation of the region of plume presence (ROPP) and the offshore SPM free rectangle used in MODIS dataset filtering. (**B**) Mean offshore water-leaving reflectances in the MODIS red band obtained in the offshore SPM free rectangle (**A**) for all cloud-free MODIS images in the year 2014. The $\rho_w$(R) values are plotted as a function of days and modeled with a Gaussian curve. MODIS images with differences between the offshore reflectances and the modeled reflectances larger than 1.5 times the standard deviation of the offshore reflectances were considered outliers and excluded from the dataset.

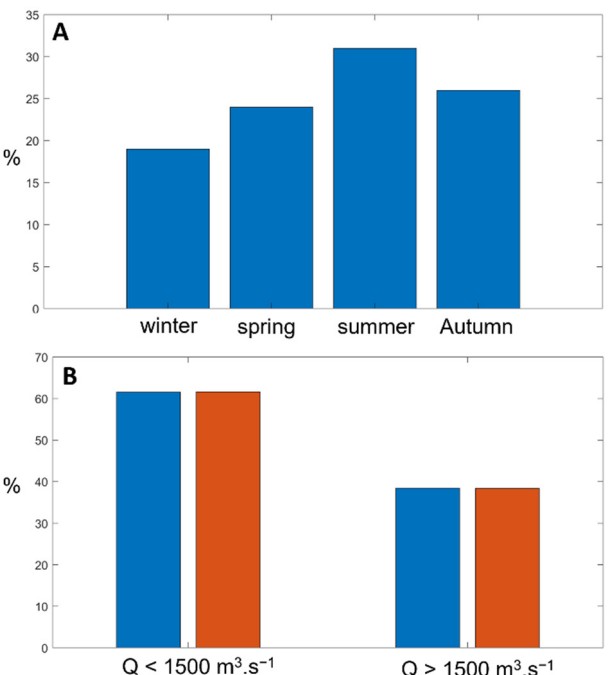

**Figure A2.** (**A**) Percentage of days covered by the MODIS dataset with respect to seasons. (**B**) Percentage of days covered by the MODIS dataset (blue bar) with dry ($Q < 1500$ m³.s⁻¹) and wet ($Q > 1500$ m³.s⁻¹) conditions compared to the outflow conditions over the 2005–2020 period (orange bar).

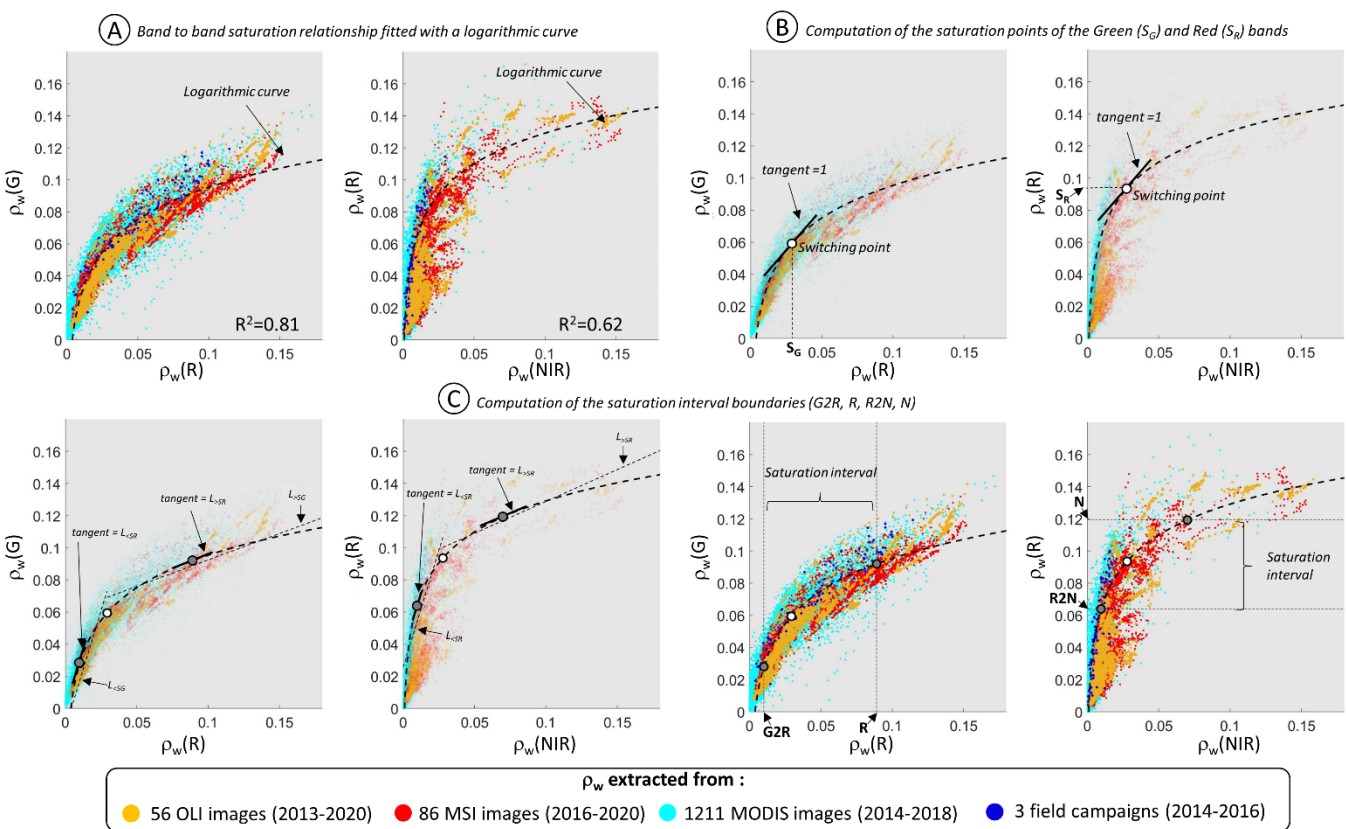

**Figure A3.** Method used to compute the switching algorithm saturation points and saturation interval radiometric bounds from band-to-band relationships. (**A**) Band-to-band relationships obtained using

$\rho_w$ data extracted from all satellite images available in the dataset (2013–2020) and measured in the field during sea campaigns (2014–2016) were fitted with a logarithmic regression curve. Satellite $\rho_w$ data were extracted along a transect from the SORA station to the offshore part of the river plume on OLI and MSI images and from the river mouth to the offshore limit of the plume on MODIS images. (**B**) Computation of the saturation points for the green ($S_G$) and red ($S_R$) bands. Saturation points are considered to be the points where the tangents of the logarithmic curves are equal to 1. (**C**) Computation of the saturation intervals where the switch between bands is performed using weighting factors that allow a smooth transition. The saturation interval radiometric bounds (G2R, R, N, and N2R) are considered to be the points where the tangent of the logarithmic curves equals the linear approximations of the logarithmic curves before (L < S) and after (L > S) the saturation points (see details in Annex A). The saturation points (SG and SR) and saturation interval radiometric bounds (G2R, R, R2N, and N) are estimated in the red band for the three satellite sensors (Table 3).

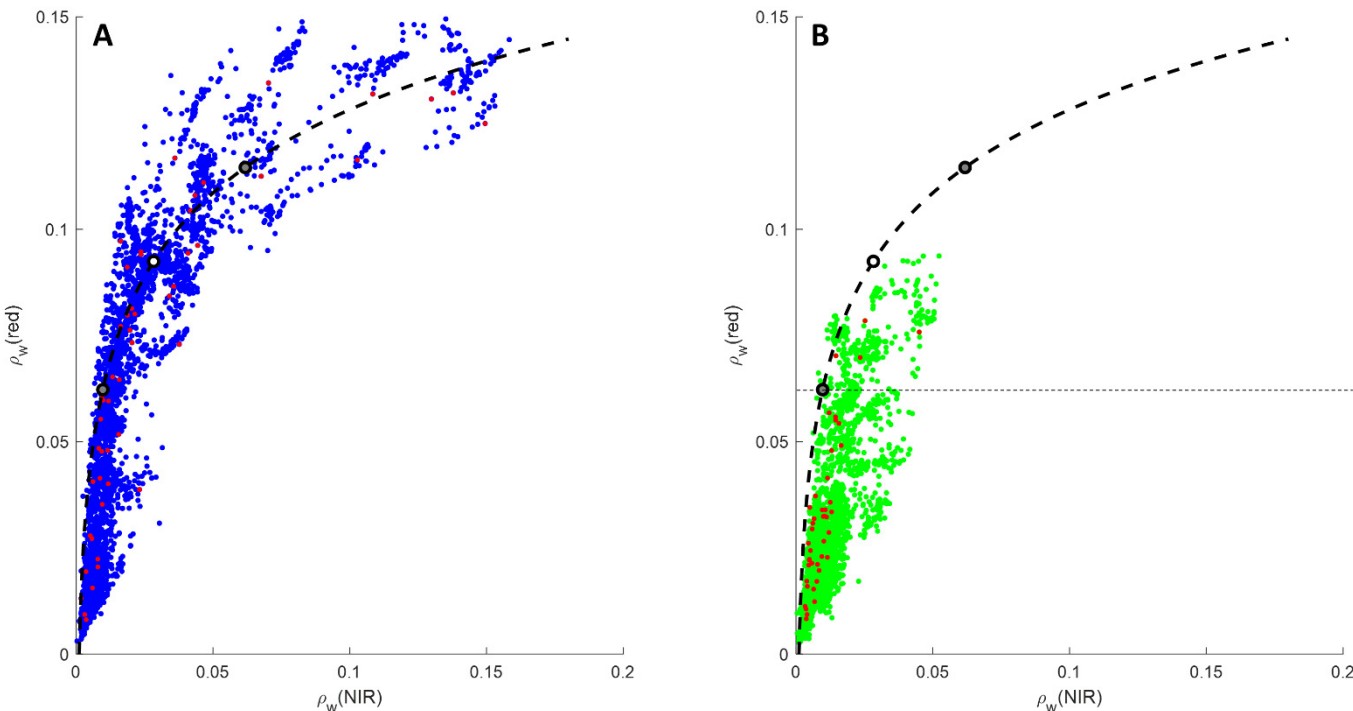

**Figure A4.** ρ(NIR) vs. ρ(R) relationships for transects extracted from OLI and MSI images (Figure A5) acquired from (**A**) October to March (blue points) and (**B**) April to September (green points). Red points correspond to the transect point extracted at the SORA station. The logarithmic curve and saturation interval boundaries computed in Figure A3 are reported. We clearly see that data points acquired in fall and winter as well as most of data points acquired at the SORA station follow the modeled logarithmic curve, while the data points acquired during spring and summer show a higher scatter due to abnormally high values of ρ(NIR), likely due to adjacency effects. Notice that SPM concentrations for data points with ρ(R) below the horizontal dashed line on B are computed using only the red band relationship and are thus weakly or not affected by this effect.

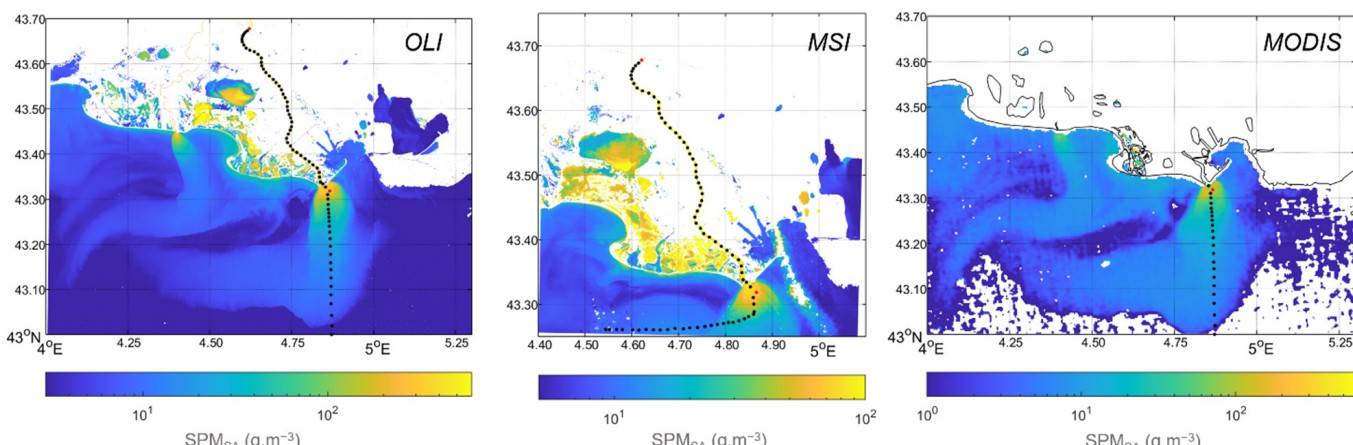

**Figure A5.** SPM concentration maps with the location of the transect points used in the study for OLI, MSI, and MODIS. The red points correspond to the SORA and MesuRho transect points. For MSI, the tiles end just after the Rhône River mouth; the transect is thus deviated towards the west in order to follow a smooth SPM concentration decrease as much as possible.

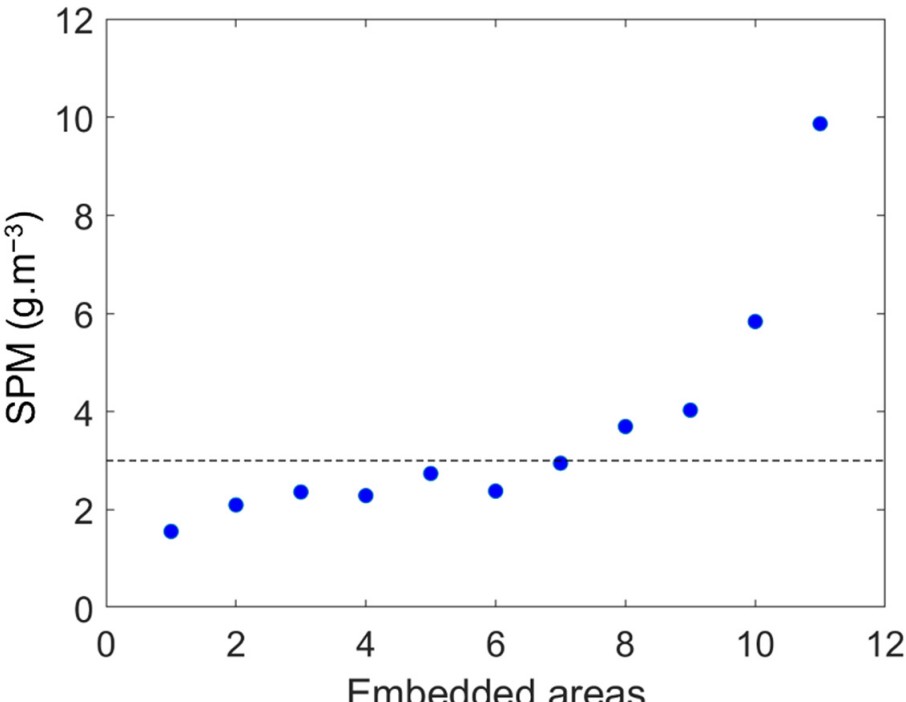

**Figure A6.** Percentile 95 values of SPM concentration (g.m$^{-3}$) for the twelve embedded areas. The shaded line corresponds to a concentration of 3 g.m$^{-3}$ and is considered to be the limit between the SPM concentration background and the SPM concentration of the turbid plume.

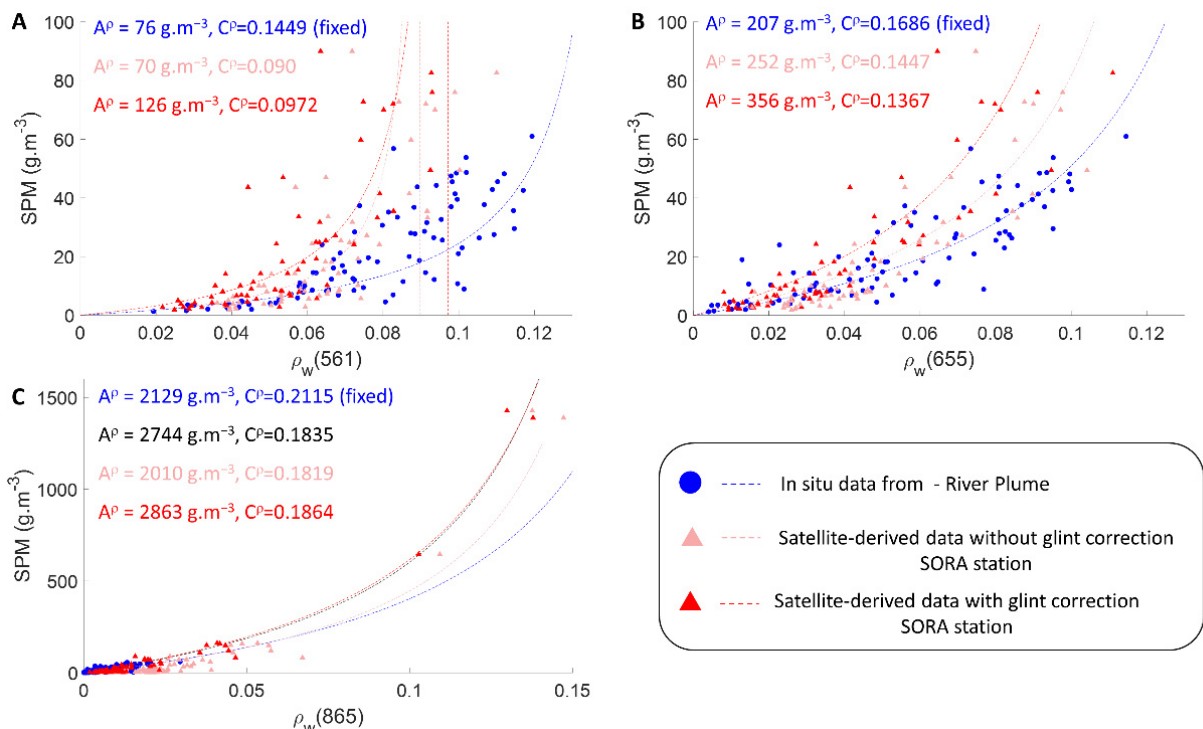

**Figure A7.** $\rho_w$ vs. SPM relationships obtained for the green (**A**), red (**B**) and NIR (**C**) bands of the OLI sensor with in situ data acquired in the Rhône River plume during sea campaigns (blue points and blue dashed lines); OLI- and MSI-derived $\rho_w$ values extracted from the SORA pixel with (red triangles and red dashed line) and without (pink triangles and pink dashed line) glint correction and in situ SPM concentration measured at the SORA station. The relationships selected in this study correspond to the blue dashed line for the green and red relationships and to the dark line for the NIR one. The $C^\rho$ coefficients are fixed to the [38] values for the river plume relationships but are fitted to the data for the satellite-derived relationship from the SORA station.

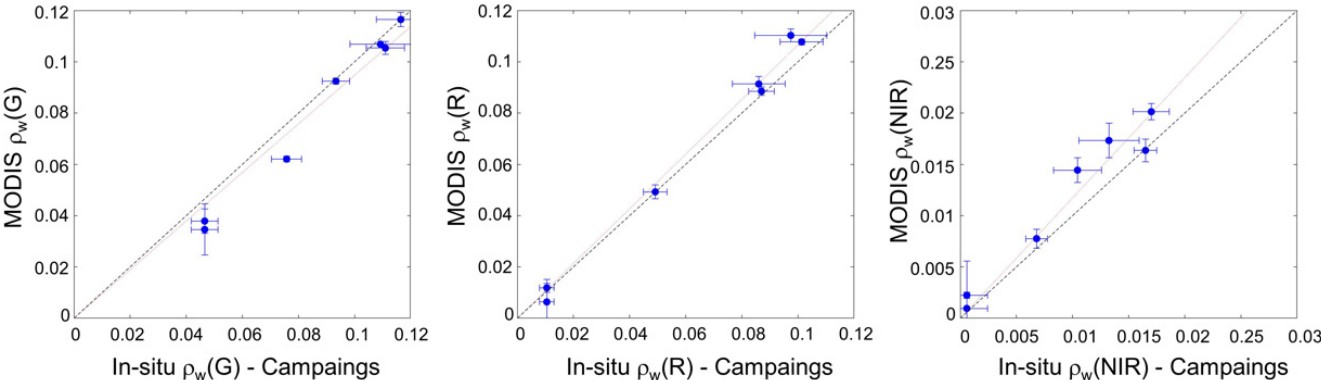

**Figure A8.** Water-leaving reflectance matchups obtained between the MODIS 250 m resolution green (555 nm), red (645 nm), and NIR (859 nm) bands and the in situ water-leaving reflectance data acquired in the river plume during sea campaigns.

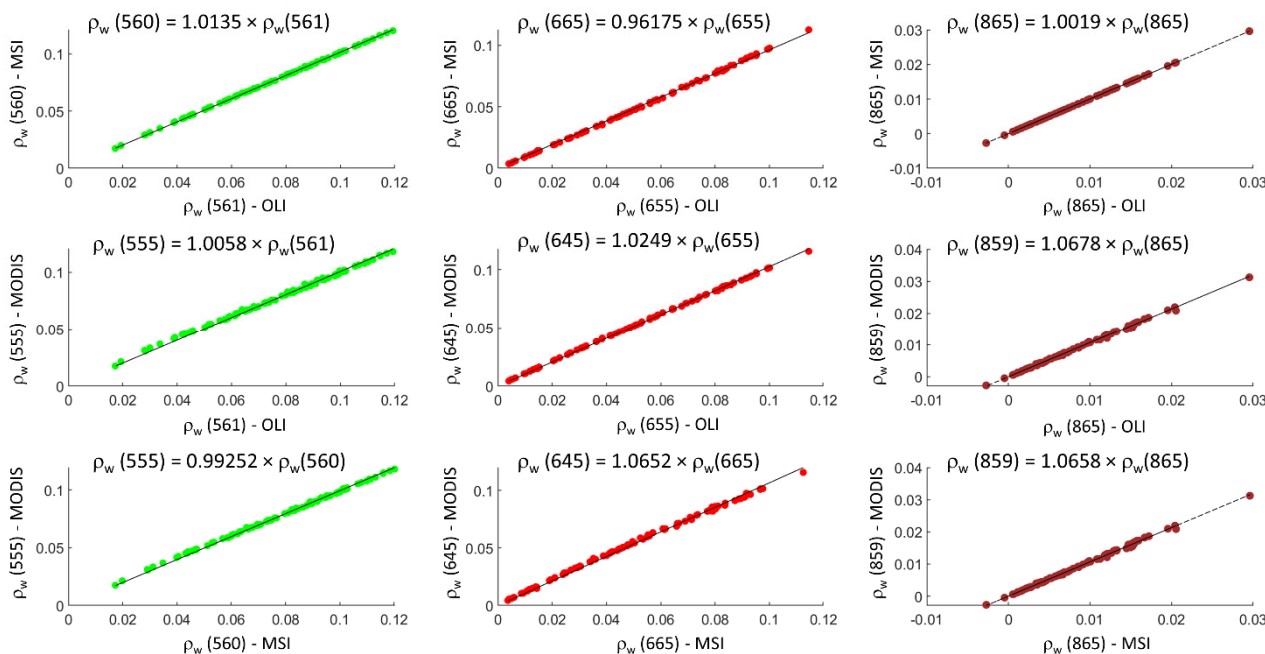

**Figure A9.** Linear relationships obtained between $\rho_w$ values in each equivalent green (green points), red (red points) and NIR (brown points) satellite sensor spectral band. Data are from field campaigns and were weighted by the spectral sensitivity of each sensor band to obtain the equivalent $\rho_w$ values. Fitted factors were used to convert $\rho_w$ values from one sensor band to another.

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
