# Peer review of "Ocean Color Remote Sensing of Suspended Sediments along a Continuum from Rivers to River Plumes: Concentration, Transport, Fluxes and Dynamics"

_remotesensing, doi:10.3390/rs14092026_

Round 1
Reviewer 1 Report
The authors propose an interesting development on the important issue of estimating river SPM concentration and mass distribution from satellite data. An upgrade of a switching algorithm to serve not only in the river plume area but also its mouth and someway upstream is profoundly elaborated and validated against other methods and ground measurements. Though demanding further consideration of limiting factors, especially in the river stream region, the new algorithm seems promising. Its application for other rivers would be a valuable follow-up of the work.
The article is clear, well-written and can be published in its present form, after maybe one more check for minor language corrections, such as:
Line 1018: experience instead of experiment
Author Response
Thank you very much for these positive comments. We have carefully checked the text for language corrections and improved the quality of Figures.
Line 1018: ‘experience’ is now used instead of ‘experiment’
Reviewer 2 Report
The author is very experienced in retrieving offshore high suspended solids concentration by ocean color remote sensing. In this paper, he skillfully combined high-resolution Landsat and medium resolution ocean color satellite data to retrieve the total suspended matter from river to estuary plume By retrieving the total suspended matter. It can provide a useful reference for the research and management decision-making of estuarine area with important economic and social impact.
(1) This paper mainly includes the inversion of suspended solids and the analysis of plume flow force in the estuary. The paper does not fully explain the complementary advantages between medium resolution and high resolution, and between medium time resolution and low time resolution. In the part of TSM inversion, the possible effects of atmospheric correction, glint and adjacency effects are analyzed, but there is a lack of sufficient quantitative analysis, especially glint and adjacency effects. The analysis of plume flow force is also insufficient. It is suggested that the paper be divided into two articles.
(2) Line 342 in the text shows that C and CP are measurable parameters related to inherent optical characteristics. At the same time, table 2 provides several cases of fitting to obtain CP. can the author supplement the comparison between the fitting results and the measured results.
(3) There are also some drafting errors in the text. For example:
(i) There are several reference errors in the text, such as "error! Reference source not found" on lines 135 and 156 It is recommended to update the reference again.
(II) table index error, such as table 1 in line 395 should be Table 2; Table 1a in row 413 should be table 3A; Figure 2.B in line 438 incorrect; 465 lines, 472 lines, 480 lines, 483 lines, and so on. It is recommended to recheck whether the index and reference of the icon are correct. "Figure 4.3-4" in line 417 cannot be found.
(III) one more "is a" in line 120;
(IV) "section 0" appears many times in the text, but it does not appear in the text.
(v) "Latitude < 4.5 ° n" in line 525 is incorrect.
(VI) what does "(10)" in line 889 mean?
Author Response
The author is very experienced in retrieving offshore high suspended solids concentration by ocean color remote sensing. In this paper, he skillfully combined high-resolution Landsat and medium resolution ocean color satellite data to retrieve the total suspended matter from river to estuary plume By retrieving the total suspended matter. It can provide a useful reference for the research and management decision-making of estuarine area with important economic and social impact.
Reply:
Thank you very much for the positive comments.
(1) This paper mainly includes the inversion of suspended solids and the analysis of plume flow force in the estuary. The paper does not fully explain the complementary advantages between medium resolution and high resolution, and between medium time resolution and low time resolution. In the part of TSM inversion, the possible effects of atmospheric correction, glint and adjacency effects are analyzed, but there is a lack of sufficient quantitative analysis, especially glint and adjacency effects. The analysis of plume flow force is also insufficient. It is suggested that the paper be divided into two articles.
Reply:
We agree the paper is long and several methodology sections are quite technical. However, we prefer to maintain the paper as one article to highlight the combined use of high spatial and medium resolution satellite observations to monitor the transport of SPM from the inside part of the river to the offshore limits of the river plume.
In order to shorten the paper, two technical sections (2.3.1 ‘Rhow vs SPM relationships’ and 2.3.2 ’Selection of switching algorithm radiometric bounds’) are now summarized while most of their technical contents was moved to ‘Supplementary Materials’. Technical sections being shorter, the Discussion section is now longer to better explain the advantages of combining high spatial resolution satellite data (MSI, OLI) to monitor the concentrations and transport of SPM inside the river (from the gauging station to the mouth) with high temporal resolution satellite data (MODIS) to estimate the resulting extent and mass of SPM within the surface river plume.
We agree that glint and adjacency effects were mentioned and considered in the study but without a sufficient quantitative analysis.
The Harmel et al. (2018) correction implemented into the ACOLITE software was used to estimate and correct for glint effects. On average this glint correction results in the decrease of the water-leaving reflectance (rhow) values by about 0.004 in the green, red and NIR bands. Considering the ranges of rhow values encountered inside the river, in these three spectral bands (see Figure S3), these impacts are therefore significant but lower than 10% of the total visible rhow signal (around 20% in the NIR). This impact in now reported in section 2.2.3 (line 271) and on the new Figure S7 (Figure S8 in the previous version). Residuals glint effects may have a significant impact on rhow values mainly in the NIR band (especially when SPM concentration is low). On the opposite, results also show that glint effects can be overcorrected in the case of high SPM concentrations. Match-ups between field radiometric measurements and satellite (MSI and OLI) data inside the river are required to better quantify these glint effect and assess the validity of the corrections applied. We rewrote several paragraphs in the match-up section (3.1) and the discussion (4.1) to better emphasize the potential impact of glint effects and glint correction errors in our results.
The SIMilarity Environment Correction (SIMEC) (Sterckx et al. 2015) can be activated in the iCOR atmospheric correction algorithm (installed as plug-in in SNAP) to estimate and correct for adjacency effects (although we do not know how valid is such correction in our study area). Anyway we processed using iCOR without then with SIMEC a selection of S2-MSI images of the Rhône River (downstream part) representing different cases: clear to turbid waters, low (fall-winter) to high (spring-summer) reflective surrounding lands. The estimated glint effects on the multi-spectral water reflectance signal were computed along a transect (15 pins) from the SORA station to the river plume (see Fig. 1) as:
Glint (%) = 100 * ((rhow_nosimec – rhow_simec) / rhow_nosimec)
For the most favorable cases of minimum glint (turbid waters surrounded by low reflective land), glint effects were estimated to be about 10% at 560 and 665 nm, 15% at 865 nm. For the worst cases of maximum glint (clear waters surrounded by highly reflective land), glint effects were estimated to be about 20%% at 560 and 665 nm, up to 80%% at 865 nm (Figure 1). Even on this last case (low turbid waters), SPM will be estimated using rhow in the green and red bands resulting in a low glint-induced effects on the retrieved SPM concentrations along the river. This is now clearly stated in the text (lines 442-453).
Figure 1. Examples of glint effects on aS2-MSI image for the worst-case scenario (clear water ad highly reflective land) estimated using the SIMEC algorithm. Results are presented for 15 pins along the river transect from SORA to the plume.
Finally, we believe the analysis of plume flow force is sufficient: whole section 3.3 and Figure 10.
(2) Line 342 in the text shows that C and CP are measurable parameters related to inherent optical characteristics. At the same time, table 2 provides several cases of fitting to obtain CP. can the author supplement the comparison between the fitting results and the measured results.
Reply: The CP parameter in the Nechad et al. (2010) equation mainly defines the saturation part of the relationship. The green and red relationships are mainly used in their linear regime (before saturation) and were built using in situ data with a SPM concentration range from 0 to 60 g.m-3 that does not allow fitting the saturation part of the relationships. Therefore, we used as default the CP parameter value calculated in Nechad et al. (2010) for the southern Norh Sea. The mass-specific inherent optical (IOP) properties used in Nechad et al. (2010) are actually close the values measured in the Rhône River plume (Lorthiois 2012) so that the Nechad CP parameter value provides a good fit for our relationships. The saturation part of the NIR relationship is used to estimate the high SPM concentration and thus needs to be well calibrated. Moreover, the absorption coefficient is poorly constrained in the NIR by the standard IOP values at it is considered to be null, while Lorthiois 2012 measured a value of 0.0084 m2.g-1 at 770 nm in the Rhone River plume. As high SPM concentration data were available to fit the NIR relationship, we thus decided to keep the CP parameter free and to fit it simultaneously with the AP parameter. The CP values obtained with the fit are lower than those estimated by Nechad et al. (2010) but are in agreement with estimations made using data from the Rhone RIver plume (Lorthiois 2012). These more detailed explanations and comparisons with measured results are now presented in the Annex S1.
(3) There are also some drafting errors in the text. For example:
(i) There are several reference errors in the text, such as "error! Reference source not found" on lines 135 and 156 It is recommended to update the reference again.
Reply: Done
(II) table index error, such as table 1 in line 395 should be Table 2; Table 1a in row 413 should be table 3A; Figure 2.B in line 438 incorrect; 465 lines, 472 lines, 480 lines, 483 lines, and so on. It is recommended to recheck whether the index and reference of the icon are correct. "Figure 4.3-4" in line 417 cannot be found.
Reply: Done
(III) one more "is a" in line 120;
Reply: Done
(IV) "section 0" appears many times in the text, but it does not appear in the text.
Reply: Done
(v) "Latitude < 4.5 ° n" in line 525 is incorrect.
Reply: Done, replaced by "Latitude < 42.5 ° n"
(VI) what does "(10)" in line 889 mean?
Reply: It was an old reference, we removed it.

Reviewer 3 Report
I have many concerns about the reviewed work, the most important of which I outline below:
- After reading this paper, I repeatedly do not know what I learned from this work? I do not know what goals the authors have set for this study? Unfortunately, the work does not verify any hypothesis. The authors wrote that their work is an extension of the previous paper, is that enough for a scientific article?
- I have doubts about the methodology used in this work. Can the SPM measurement at a distance of 45 km be reliable? After all, it is proven that both SPM and turbidity varies for almost every pixel considered. Please see any paper related to the topic of this study. I am not convinced by this methodology.
The paper contains a huge number of editorial errors that make it difficult to read, for example:
- Many figures are illegible or have too small font (i.e. Figs. 3, 63 S8.)
- The paper contains many errors in literature references, tables, and figures. Probably due to formatting in LaTeX.
- The multiplication sign in LaTeX is \cdot Many errors appeared in the paper because of this.
- The paper is difficult to read due to its length. It is the authors' responsibility to write concisely and to the point.
The work also contains many colloquialisms, e.g.
- In the line 41 the authors wrote: "Most" I would like to know exactly how many percent? Based on what sources they claim this.
- In the line 45 the authors wrote: "these gauging stations are usually located up to 100km". Which ones exactly? What does "usually" mean? In a scientific paper you should give exact numbers and be specific.
- The paragraph from line 55 to 60 has no reference. Citation is essential.
Similar errors could be identified at the whole paper.
Author Response
I have many concerns about the reviewed work, the most important of which I outline below:
After reading this paper, I repeatedly do not know what I learned from this work? I do not know what goals the authors have set for this study? Unfortunately, the work does not verify any hypothesis. The authors wrote that their work is an extension of the previous paper, is that enough for a scientific article?
Reply:
As clearly stated in the text, we believe our study is the first one combining high and medium spatial resolutions satellite data to monitor SPM concentrations along a river, river mouth and river plume continuum. An existing SPM switching algorithm (Novoa et al. 2017) developed for estuaries has been adapted to this type of continuum and SPM retrievals were validated using in situ data from the river gauging data. Moreover, our results go further than a previous study (Gangloff et al. 2017) by relating SPM concentrations (and estimated masses) at the river gauging station, river mouth and within the surface river plume. We believe our study is definitely original enough to justify a publication, as acknowledged by two other reviewers. Several sentences in Introduction (lines 76-78) and Conclusion (lines 1026-1029) were modified to better highlight the originality of our work.
I have doubts about the methodology used in this work. Can the SPM measurement at a distance of 45 km be reliable? After all, it is proven that both SPM and turbidity varies for almost every pixel considered. Please see any paper related to the topic of this study. I am not convinced by this methodology.
Reply:
We are not sure to fully understand the reviewer comment: in situ measurements made at the Rhône River gauging station (45 km upstream the river mouth) are reliable. During flood events, SPM concentrations are measured in the field every 4 hours to consider the temporal variations. One of the questions addressed in our study was actually: can we accurately retrieve SPM concentrations along the river (including at the river gauging station) using high spatial resolution ocean color satellite data. The answer to this question is presented in our Figure 7: using the switching algorithm, and despite errors associated to imperfect atmospheric, glint and adjacency effects corrections, SPM concentrations ranging from 60 to 1600 g.m-3 are retrieved with a RMSE of 37 g.m-3, which is a satisfactory result in the scope of our study.
We added sentences in the match-up section (3.1, lines 629-638) and the discussion section (4.1, lines 970-975 in the discussion) to report on the impact of spatial and temporal variations of the SPM concentration on the comparison between satellite-derived and in situ SPM concentrations (match-up quality).
The paper contains a huge number of editorial errors that make it difficult to read, for example:
Many figures are illegible or have too small font (i.e. Figs. 3, 63 S8
The paper contains many errors in literature references, tables, and figures. Probably due to formatting in LaTeX.
The multiplication sign in LaTeX is \cdot Many errors appeared in the paper because of this.
Reply:
Done: Figures 3, 7, S8 (now S7) and S3 (now S9) were modified with an increased font size.
All reference errors have been corrected, we apologize for this.
The paper is difficult to read due to its length. It is the authors' responsibility to write concisely and to the point.
Reply:
We want to apologize for the syntax errors in the manuscript, notably the errors in literature references, tables, and, which occurred during the docx to pdf conversion of the file. All these editorial errors have now been corrected for in the revised version. We also apologize for the legibility of several figures too small font; this issue also was carefully rectified.
We agree the paper is long and includes some very technical sections. In order to shorten the paper, two technical sections (2.3.1 ‘Rhow vs SPM relationships’ and 2.3.2 ’Selection of switching algorithm radiometric bounds’) were now summarized while their technical contents were moved in Supplementary Materials.
The work also contains many colloquialisms, e.g.
In the line 41 the authors wrote: "Most" I would like to know exactly how many percent? Based on what sources they claim this.
In the line 45 the authors wrote: "these gauging stations are usually located up to 100km". Which ones exactly? What does "usually" mean? In a scientific paper you should give exact numbers and be specific.
Reply: “Most”: based on the reviewer comment, we replaced the word ‘most’ by a more appropriate ‘many’ as it is easy to list hundreds of rivers nowadays equipped with gauging stations.
Examples of distance between these gauging stations and the river mouths:
- Obidos (Brazil) is located >200km upstream the Amazon River mouth.
- Tsiigehtchic (Canada) is located about 100 km upstream the Mackenzie River mouth.
- Kyusyur (Russia) is located about 120 km upstream the Lena River mouth.
Actually, most of the gauging stations of major Arctic rivers are located about 100 km upstream the actual river mouths (source: https://arcticgreatrivers.org/rivers/).
Based on the reviewer’ question we replaced the word "usually" by “often”
The paragraph from line 55 to 60 has no reference. Citation is essential.
Similar errors could be identified at the whole paper.
Reply:
We agree with this comment and now refer in this paragraph to four studies relying on autonomous monitoring networks in rivers and estuaries:
- Eyrolle F., Lepage H., Antonelli C., Morereau A., Cossonnet C., Boyer P., Gurriaran R. (2020).
- Radionuclides in waters and suspended sediments in the Rhone River (France) - Current contents, anthropic pressures and trajectories. Science of The Total Environment,723, https://doi.org/10.1016/j.scitotenv.2020.137873.
- Jalon-Rojas I., Schmidt S., Sottolichio A. (2017) Comparison of environmental forcings affecting suspended sediments variability in two macrotidal, highly-turbid estuaries, Estuarine, Coastal and Shelf Science, doi: 10.1016/j.ecss.2017.02.017.
- Jalon-Rojas I., Sottolichio A., Hanquiez V., Fort A., Schmidt S. (2018) To what extent multidecadal changes in morphology and fluvial discharge impact tide in a convergent (turbid) tidal river, Journal of Geophysical Research - Oceans, 123, 3241-3258, doi: 10.1002/2017JC013466.
- Martinez J.M., Guyot J.L., Naziano F., F. Sondag (2009. Increase in suspended sediment discharge of the Amazon River assessed by monitoring network and satellite data. CATENA, 2009, 29.

Round 2
Reviewer 3 Report
I agree with the authors' responses, which convince me. However, I miss a reference more supported by the literature regarding my earlier comment "After all, it is proven that both SPM and turbidity varies for almost every pixel considered." I would suggest, for example, looking at a paper:
Investigation of Sediment-Rich glacial meltwater plumes using a high-resolution multispectral sensor mounted on an unmanned aerial vehicle by Wójcik et al., Water 11 (11), 2405, 2019.
where it is shown how turbidity can vary significantly, even at very small distances between points.
Thus, it would be good for the paper to add a paragraph on the limitations of the method and algorithm used. If this comment is clarified, I think the paper can be published.
Author Response
Reply
We agree with the reviewer comment, i.e. comparison between satellite-derived values at 20 m spatial resolution and field measurements on a fixed point in turbid and dynamic waters is an issue. This was highlighted by Wójcik et al. (2019) using 15 cm spatial resolution remote sensing data.
One paragraph in the Results section was modified according to this comment (lines 625-639) and two new references were added to support it.
Modification (in yellow) in the manuscript
Lines 625-639
The spatial variability of SPM concentrations in turbid and dynamic waters can be an issue when comparing satellite-derived values with field measurements on a fixed point, even for high (meter scale, e.g. Luo 2020) to very high (centimeter scale, e.g. Wójcik et al. (2019)) resolution sensors. The impact of the SPM concentration spatial variability was thus estimated on these match-ups by converting the rhow standard deviation computed in the 3x3 satellite pixels box around the SORA station into SPM concentration standard deviation (error bars on Figure 7A, which actually do not appear being lower than the marker size). This impact is limited with a spatial variability lower than ± 15% which is consistent with Luo et al. (2020) who observed a spatial variability of water turbidity lower than 10% when downscaling from to 2 to 20 m Pleiades satellite data recorded over highly turbid estuarine waters. This impact is also minimized by the number of field measurements used to compute the daily-averaged SPM concentration at the SORA station (up to 6 measurements a day during flood conditions), which allows to smooth the turbidity small scale variability. The day-to-day SPM concentration variability at the SORA station is higher, with some data points showing variability larger than 50 %. However, removing these data points only slightly improved the mean residual
error (33 g.m-3) also suggesting a limited impact.
